# Investigation of Aerosol-Cloud Interactions under Different Absorptive Aerosol Regimes using ARM SGP Ground-Based Measurements

Xiaojian Zheng[1], Baike Xi[1], Xiquan Dong[1], Timothy Logan[2], Yuan Wang[3,4] and Peng Wu[1]

[1]Department of Hydrology and Atmospheric Sciences, University of Arizona, Tucson, AZ, USA

[2]Department of Atmospheric Sciences, Texas A&M University, College Station, TX, USA

[3]Division of Geological and Planetary Sciences, California Institute of Technology, Pasadena, CA, USA

[4]Jet Propulsion Laboratory, California Institute of Technology, Pasadena, CA, USA

*Correspondence to*: Baike Xi (baikex@email.arizona.edu)

15 **Abstract**

The aerosol indirect effect on cloud microphysical and radiative properties is one of the largest uncertainties in climate simulations. In order to investigate the aerosol-cloud interactions, a total of 16 low-level stratus cloud cases under daytime coupled boundary layer conditions are selected over the Southern Great Plains region of the United States (SGP). The physicochemical properties of aerosols and their impacts on cloud microphysical properties are examined using data collected from the Department of Energy Atmospheric Radiation Measurement (ARM) facility at SGP site (ARM-SGP). The aerosol-cloud interaction index ($ACI_r$) is used to quantify the aerosol impacts with respect to cloud-droplet effective radius. The mean value of $ACI_r$ calculated from all selected samples is $0.145 \pm 0.05$ and ranges from 0.09 to 0.24 at a range of cloud liquid water paths (LWP=20-300 g m$^{-2}$). The magnitude of $ACI_r$ decreases with increasing LWP, which suggests a diminished cloud microphysical response to aerosol loading presumably due to enhanced condensational growth processes and enlarged particle size. The impact of aerosols with different light-absorbing abilities on the sensitivity of cloud microphysical responses is also investigated. In the presence of weak light-absorbing

aerosols, the low-level clouds feature a higher number concentration of cloud condensation nuclei ($N_{CCN}$) and smaller effective radii ($r_e$) while the opposite is true for strong light-absorbing aerosols. Furthermore, the mean activation ratio of aerosols to CCN ($N_{CCN}/N_a$) for weakly (strongly) absorbing aerosols is 0.54 (0.45), owing to the aerosol microphysical effects, particularly the different aerosol compositions inferred by their absorptive properties. In terms of the sensitivity of cloud droplet number concentration ($N_d$) to $N_{CCN}$, the fraction of CCN that converted to cloud droplets ($N_d/N_{CCN}$) for the weakly (strongly) absorptive regime is 0.69 (0.54). The measured $ACI_r$ values in the weakly absorptive regime are relatively higher, indicating that clouds have greater microphysical responses to aerosols, owing to the favorable thermodynamic condition. The reduced $ACI_r$s in the strongly absorptive regime are due to the cloud-layer heating effect induced by strong light-absorbing aerosols. Consequently, we expect larger shortwave radiative cooling effects from clouds in the weakly absorptive regime than those in the strongly absorptive regime.

## 1. Introduction

Clouds play a critical role in the Earth's climate by acting as the dominant modulator of radiative transfer in the atmosphere and have substantial impacts on the global climate. The radiative effect of clouds contributes to one of the largest uncertainties in climate modeling (IPCC, 2013), and has been well known to be influenced by aerosol loading. An increase in aerosol concentration can lead to the enhancement of cloud droplet number concentration ($N_d$) and the reduction of cloud droplet effective radii ($r_e$), which results in an increase of cloud albedo. This phenomenon is defined as the aerosol first indirect effect (Twomey, 1977), and it is denoted as a general cooling effect in terms of global radiation balance. More fundamentally, the aerosol effects on cloud reflectance result from the cloud microphysical response to aerosol concentration (e.g., aerosol-cloud interaction, ACI).

The magnitude and sensitivity of ACIs in low-level clouds have been investigated by numerous studies, using various observational datasets such as ground-based measurements (Garrett et al., 2004; Feingold et al., 2006; Kim et al., 2008; McComiskey et al., 2009; Wang et al., 2013, 2018a), satellite retrieved products (Sekiguchi et al., 2003; Su et al., 2010) and

airborne in situ measurements (Twohy et al., 2013; Painemal and Zuidema, 2013; Zhao et al., 2018). However, large variations exist among various assessments, because of intrinsic instrument uncertainty, differing analysis methods, and more physically, the inherent variation in aerosol properties. The physical mechanism underlying the aerosol effect on clouds is that aerosols activate as cloud condensation nuclei (CCN) and then influence the cloud microphysical features. The efficacy of the activation of aerosol has been widely known to be influenced by aerosol size distribution and chemical composition which are the primary sources of uncertainty in assessing ACI (Dusek et al., 2006; McFiggans et al., 2006; Liu and Li, 2014; Che et al., 2016).

Previous studies have suggested that the composition of aerosols can be inferred by their optical properties such as aerosol optical depth, single scattering albedo, and Ångström exponent (Clarke et al., 2004; Bergstrom et al., 2007; Clark et al., 2007; Russell et al., 2010; Cazorla et al., 2013; Cappa et al., 2016). For instance, fine mode carbonaceous particles (e.g., black and organic carbon) have strong light-absorbing abilities in the ultraviolet and visible spectra (Logan et al., 2013). Urban pollution aerosols associated with sulfate and nitrate particles are considered as weakly absorbing aerosols (Eck et al., 1999, 2005; Bergstrom et al., 2007; Chin et al., 2009). Although studies have been done to classify aerosol types using the absorption Ångström exponent, which is associated with the absorptive spectral dependence of particles, the measurements of this parameter typically carry large uncertainty, and can provide limited information when there are mixtures of different aerosol species that share similar spectral dependences (Bergstrom et al., 2007; Lack and Cappa, 2010). Alternatively, the single scattering albedo (SSA) and co-albedo (1-SSA) can be used to better separate the aerosol types because they focus on the relative absorbing ability of aerosols at specific wavelengths (Logan et al., 2013; Tian et al., 2017). Given the wide availability of aerosol optical property measurements, the feasibility of inferring aerosol species from their optical properties is useful particularly in areas with no direct measurements of aerosol chemical composition (Logan et al., 2013; Schmeisser et al., 2017).

The Atmospheric Radiation Measurement (ARM) program initiated by the U. S. Department of Energy (DOE) aims to improve the parameterization of clouds in global climate

models (Stokes and Schwartz, 1994). Thus far, the ARM program has established over 20 years of long-term ground-based measurements of cloud properties and surface measured aerosol properties at the Southern Great Plain (SGP) site which represents typical continental conditions (Ackerman and Stokes, 2003; Dong et al., 2005). The size and composition of aerosols have been found to have a considerable seasonal and regional dependence, and their impacts on clouds also vary with different aerosol regimes (Sorooshian et al., 2010; Logan et al., 2018). The prevailing fine mode aerosols at the ARM-SGP site typically contain organic and black carbon associated with biomass burning and inorganic aerosols composed of sulfate and nitrate species (Parworth et al., 2015; Logan et al., 2018). The differences in intrinsic hygroscopicity among those aerosol species play various roles in aerosol activation processes and consequently lead to various interactions with clouds. Thus, it is necessary to investigate the aerosol and cloud properties as well as the magnitude of the ACI index at the ARM-SGP site, in order to (a) enhance the understanding of ACI and (b) reduce the uncertainty in quantifying the ACI and associated radiative effects when modeling aerosol influences on low-level continental clouds.

In this study, the aerosol and cloud properties at the ARM-SGP site from 16 selected non-precipitating low-level stratiform cloud cases during the 2007-2012 period are examined. Details of the observational measurement platforms and methods are introduced in section 2. The development and analysis of the ACI for the 16 selected cases, the aerosol activation and cloud microphysical responses, as well as consequent cloud radiative effects under different aerosol absorptive regimes, are investigated in section 3. Lastly, a summary of our findings and future work is presented in section 4.

## 2    Data and methods

### 2.1    Cloud Properties

#### 2.1.1    Cloud Boundaries

The cloud boundaries at the ARM-SGP site were primarily determined by the ARM Active Remotely-Sensed Cloud Locations (ARSCL) product, which is a combination of data detected by multiple active remote-sensing instruments, in particular, the Millimeter-wavelength Cloud

Radar (MMCR). The MMCR operates at a frequency of 35 GHz (and wavelength of 8.7 mm) with a zenith pointing beamwidth of 0.2° and provides a continuous time-height profile of radar reflectivity with temporal and spatial resolutions of 10 seconds and 45 m, respectively (Clothiaux et al., 2000). After 2011, the MMCR was replaced by the Ka-band ARM Zenith Radar (KAZR) which has the same operating frequency and shares similar capabilities as the MMCR, but with the major improvement of a new receiver that allows for more sensitivity in cloud detection (Widener et al., 2012). The temporal and vertical resolutions of KAZR-detected reflectivity are 4 seconds and 30 m, respectively. The cloudy condition, as well as cloud top height, is identified via cloud radar reflectivity. The uncertainties of cloud top height detected by MMCR and KAZR are 45 m and 30 m, respectively.

The cloud radar is sensitive to the sixth moment of droplet size distribution and can be contaminated by insects below the cloud base (Dong et al., 2006). The laser ceilometer measurement, which is sensitive to the second moment, is used to provide an accurate cloud base estimation. The uncertainty of cloud base height is around 10 m (Morris, 2016). Hence, the lidar-radar pair provides the most precise determination of cloud boundaries from a point-based perspective, with combined uncertainties of cloud thickness for MMCR and KAZR periods are 55 m and 40 m, respectively. Note that this will not cause a significant difference in determining the cloud boundaries between these two radar periods. In this study, the cloud base and top heights were averaged into 5-min bins, where the low-level stratus cloud is defined as a cloud-top height lower than 3 km with no overlying cloud layer (Xi et al., 2010).

### 2.1.2 Cloud Microphysical Properties

The cloud liquid water path (LWP), defined as the column-integrated cloud liquid water, was retrieved based on the measured brightness temperatures from the Microwave Radiometer (MWR) at 23.8 and 31.4 GHz, using the statistical method described in Liljegren et al. (2001). The uncertainty of LWP retrieval is 20 $g\,m^{-2}$ for LWP less than 200 $g\,m^{-2}$ and around 10% for LWP higher than 200 $g\,m^{-2}$. In this study, we exclude the data points with LWPs less than 20 $g\,m^{-2}$ to eliminate optically thin clouds, as well as exclude the samples with LWPs greater than 300 $g\,m^{-2}$ to prevent potential precipitation contamination issues (Dong et al., 2008).

For microphysical properties of low-level stratus, following the methods developed by Dong et al. (1998), the daytime layer-mean cloud-droplet effective radius ($r_e$) can be calculated by:

$$r_e = -2.07 + 2.49\text{LWP} + 10.25\gamma - 0.25\mu_0 + 20.28\text{LWP}\gamma - 3.14\text{LWP}\mu_0, \tag{1}$$

where $\gamma$ is the solar transmission, $\mu_0$ is the cosine of solar zenith angle, and the units of $r_e$ and LWP are $\mu m$ and $100 \text{ g m}^{-2}$, respectively. $N_d$ is obtained after $r_e$ is known, by the following calculation:

$$N_d = \left(\frac{3\text{LWP}}{4\pi\rho_w r_e^3 \Delta Z}\right) \exp(3\sigma_x^2), \tag{2}$$

where $N_d$ has units of $\text{cm}^{-3}$, $\Delta Z$ is cloud thickness determined from cloud boundaries with units of m, and $\sigma_x$ is the width of the lognormal size distribution of cloud droplets, which is assumed to be a constant value of 0.38 (Miles et al., 2000). The sensitivities of retrieved $r_e$ and $N_d$ to the uncertainties of cloud LWP, $\sigma_x$ and $\gamma$ have been investigated in Dong et al. (1997 and 1998). The uncertainties of retrieved $r_e$ and $N_d$ have been estimated against aircraft in situ measurements over the ARM-SGP site (Dong et al., 2002, 2003) and other regions (Dong et al. (1998). As a result, the 10% change in cloud LWP and downward SW at the surface would cause a 10% uncertainty in $r_e$ retrieval. In addition, the $N_d$ uncertainty is statistically estimated to be 25%, compared with the aircraft in situ measurements at the Pennsylvania State University surface site during Fall 1996 (Dong et al. 1998) and at the ARM SGP site during March 2000 Cloud Intensive Observational Period (IOP) (Dong et al. 2002; Dong and Mace, 2003).

## 2.2 Aerosol Properties

Surface aerosol properties were collected from the Aerosol Observing System (AOS), a platform consisting of an array of instruments to monitor real-time aerosol information. The total condensation nuclei number concentration ($N_a$), which represents the overall loading of aerosol particles with diameters larger than 10 nm, was obtained by the TSI model 3010 condensation particle counter. The aerosol scattering coefficient ($\sigma_{sp}$) was measured by the TSI model 3653 nephelometer at three wavelengths: 450, 500, and 700 nm. The relative humidity inside the nephelometer was set to 40% to maintain a dry condition and prevent potential

aerosol hygroscopic effects (Jefferson, 2011). Moreover, the quality of retrievals has been assured using the Anderson and Ogren (1998) method. The absorption coefficient ($\sigma_{ap}$) was measured by the Radiance Research particle soot absorption photometer (PSAP) at three slightly different wavelengths (470, 528, and 660 nm), with the calibration and quality control process done by the method developed in Anderson et al. (1999). Note that both the nephelometer and PSAP employ two impactors with size cuts of 1 μm and 10 μm. The measurements switch between total aerosol (<10 μm) and submicron aerosol (<1 μm) every hour. In this study, the sub-10 μm aerosol optical properties with original 1-min temporal resolution were averaged into 5-min bins to match the cloud microphysical properties.

The optical particle counter developed by Droplet Measurement Technologies is used to measure the CCN number concentration ($N_{CCN}$). The supersaturation (SS) level inside the instrument cycles between 0.15% and 1.15% every hour. The CCN activity can be presented as a function of SS: $N_{CCN} = cSS^k$ (Twomey, 1959), where c and k are calculated by using a power-law fit for each hour. In this study, 0.2% is used as this represents typical supersaturation conditions of low-level stratus clouds (Hudson and Noble, 2013; Logan et al., 2014; Logan et al., 2018).

### 2.3   Boundary Layer Condition and Lower Tropospheric Stability

Given the fact that the aerosol properties were measured at the surface, there is a question of whether surface aerosols can be linked to what actually happens in clouds aloft. This study adopts the method presented in Dong et al. (2015), which defined the boundary layer condition into two categories: coupled and decoupled. The vertical sounding profiles at a 1-min temporal resolution were collected from the ARM Merged Sounding product with a vertical resolution of 20 m below 3 km (Mace et al., 2006; Troyan, 2012). The vertical profiles of liquid water potential temperature ($\theta_L$) and total water mixing ratio ($q_t$) for coupled and decoupled boundary layer conditions, as well as the criteria to differentiate between them, are illustrated in Fig. 1. The coupled condition was identified by the change of $\theta_L$ and $q_t$ from surface layer to cloud base of less than 0.5 K and 0.5 g/kg, respectively. These thresholds are the same as in Dong et al. (2015), originally suggested by Jones et al. (2011). In that case, the boundary layer is considered to be well-mixed and suggests that the surface aerosols are comparable to in-

cloud aerosols. In the decoupled condition, the $\theta_L$ and $q_t$ vary more drastically from surface to cloud base under decoupled conditions, which denotes a stratification of the sub-cloud layer, thereby disconnecting the surface aerosols from the ones aloft.

A study was conducted by Delle Monache et al. (2004) used in situ aerosol measurements from 59 flights from March 2000 – March 2001 to compare with the surface aerosol measurements. Their results showed that the aerosol extensive properties such as the total extinction by particles measured within the well-mixed boundary layer were well-correlated with surface measurements ($R^2$ value of 0.88). Therefore, selecting cloud cases under coupled conditions can better constrain the thermodynamic condition since the measured surface aerosols are representative in terms of aerosol-cloud interaction.

The Lower Tropospheric Stability (LTS), which is defined as the potential temperature difference between the surface and 700 hPa, is used to represent the large-scale thermodynamic condition. The LTS is obtained from the ECMWF model output, which specifically provides for analysis at the ARM SGP site. The value is obtained by averaging over a grid box of 0.56×0.56°, which is centered at SGP. The original temporal resolution of LTS is 1-hour and is then interpolated to 5-min to match the other variables, assuming the large-scale forcing would not have significant changes during every 1-hour window.

## 2.4 Shortwave radiation fluxes at the Surface

The surface measured broadband downwelling shortwave (SW) radiation fluxes and estimated clear-sky SW fluxes were collected from Radiative Flux Analysis Value Added Products (Long and Ackerman, 2000; Long and Turner, 2008), with an uncertainty of $10\,\mathrm{W\,m^{-2}}$. The combination of cloudy and clear-sky SW fluxes was used to calculate the cloud radiative effect. In order to minimize the influence of non-cloud factors, such as solar zenith angle and surface albedo, a representation of the relative cloud radiative effect (rCRE) is defined as

$$\mathrm{rCRE} = 1 - \mathrm{SW}_{\mathrm{cld}}^{dn}/\mathrm{SW}_{\mathrm{clr}}^{\mathrm{dn}}, \tag{3}$$

where $\mathrm{SW}_{\mathrm{cld}}^{dn}$ and $\mathrm{SW}_{\mathrm{clr}}^{\mathrm{dn}}$ are cloudy and clear-sky downwelling shortwave radiation fluxes, respectively (Betts and Viterbo, 2005; Vavrus, 2006; Liu et al., 2011).

## 2.5   Selection of low-level stratus cloud cases

As previously discussed, the selection of cloud cases is limited by the following criteria: non-precipitating and cloud-top height less than 3 km with lifetime more than 3 hours under the limitation of 20 g m$^{-2}$ < LWP < 300 g m$^{-2}$ and the coupled boundary layer conditions. Only daytime cloudy periods were considered in this study because the $r_e$ retrieval required the information of solar transmission (Dong et al., 1998). Note that all the variables used in the study are averaged in 5-min temporal resolution bins. A total of 16 cases were selected during the 6-year period from 2007 to 2012, which represents a total of 693 samples (~58 hours) in this study. The detailed period and the number of sample points of each case are listed in Table 1. Most cases occurred during the winter and spring months since low-level cloud occurrences are higher during those seasons (Dong et al., 2006). The 72-hour NOAA HYSPLIT backward trajectories (Stein et al., 2015) for sub-cloud air parcels that advected over the ARM-SGP site are used to identify the aerosol source regions (Logan et al., 2018). Aerosol plumes consisting of different species from local sources and long-range transport can impact the ARM-SGP site because of different transport pathways and can induce different cloud responses, which are further investigated in this study.

## 3   Result and Discussion

### 3.1   Aerosol and cloud properties of selected cases

The probability density functions (PDFs) of aerosol and cloud properties from all 16 cases are shown in Fig. 2, note that the distributions include each of the 5-min data points. For the aerosol properties shown in the top panel, the Ångström Exponent (AE) was calculated based on the nephelometer observed spectral scattering coefficient ($\sigma_{sp}$) at 450 nm and 700 nm, using the equation of $AE_{450-700nm} = -\log{(\sigma_{sp450}/\sigma_{sp700})}/\log(450/700)$. The negative log-log slope denotes the relative wavelength dependence of particle optical properties due to differences in particle sizes (Schuster et al., 2006). Therefore, AE can be a good indicator of aerosol particle sizes since AE > 1 indicates the particle size distributions dominated by fine mode aerosols (submicron), while AE < 1 denotes the dominance of coarse mode aerosols (Gobbi et al., 2007; Logan et al., 2010). The aerosol Fine Mode Fraction (FMF) is given by the

ratio, $\sigma_{\mathrm{sp1}}/\sigma_{sp10}$, where $\sigma_{\mathrm{sp1}}$ and $\sigma_{sp10}$ are the nephelometer measured scattering coefficients at 550 nm for fine mode aerosols (1 µm size cut) and total aerosols (10 µm size cut), respectively. This ratio indicates the dominant influence of fine mode aerosols owing to the physical properties of the entire aerosol plume. For example, FMF values greater than 0.6 represent the dominance of fine mode aerosol in the total population, and values less than 0.2 represent the dominance of coarse mode aerosols in the total population (Anderson et al., 2003). As illustrated in Figs. 2b and 2c, fine mode aerosols are dominant in the 16 selected cases. All AE values are higher than 1, with most of the values ranging from 1.5 to 2. In addition, the majority of the FMF values are greater than 0.6 and range from 0.7 to 0.9.

The variation in aerosol single scattering albedo (SSA) suggests different roles of the fine mode aerosol absorptive properties that influence total light extinction, which in turn is a result of different aerosol species in the plume. This is further explained in section 3.3. The distributions of $N_a$, $N_{\mathrm{CCN}}$, and $N_d$ represent typical continental aerosol conditions with mean values of $1060$ $\mathrm{cm}^{-3}$, $475$ $\mathrm{cm}^{-3}$, and $297$ $\mathrm{cm}^{-3}$, respectively, and $r_e$ values are more normally distributed with the majority of values between 7-9 µm. Note that the variation in the PDF of LWP is relatively small, which allows for a better investigation of the LWP dependence of cloud microphysical properties.

### 3.2 Measured Aerosol-Cloud-Interaction

To examine the microphysical response of cloud to aerosol loading, the quantitative Aerosol-Cloud-Interaction (ACI) term can be expressed as

$$\mathrm{ACI_r} = -\left.\frac{\partial \ln{(r_e)}}{\partial \ln{(\alpha)}}\right|_{\mathrm{LWP}}, \tag{4}$$

where $\alpha$ denotes aerosol loading. $\mathrm{ACI_r}$ represents the relative change of layer mean $r_e$ with respect to the relative change of aerosol loading, thereby emphasizing the sensitivity of the cloud microphysical response (Feingold et al., 2003; Garrett et al., 2004). Note that values of $\mathrm{ACI_r}$ have theoretical boundaries of 0-0.33, where the lower bound means no change of cloud microphysical properties with aerosol loading, and the upper bound indicates a linear relationship.

As suggested by previous studies, the $ACI_r$ should be calculated and compared at constant LWP owing to the dependence of $r_e$ on LWP (Twomey et al., 1977; Feingold et al., 2003). Therefore, in this study, we use six LWP bins ranging from 0-300 $g\,m^{-2}$ with bin size of 50 $g\,m^{-2}$ and then group the sample data accordingly. Note that the first bin is actually 20-50 $g\,m^{-2}$ due to the elimination of LWP less than 20 $g\,m^{-2}$. The $r_e$-$N_{CCN}$ relationship is presented in Fig. 3a, where only the samples from three LWP bins are used to illustrate the $r_e$-$N_{CCN}$ response. In general, $r_e$ decreases with increasing CCN number concentration as expected. The $ACI_r$ values range from $0.09 - 0.24$ with a mean value of $0.145 \pm 0.05$, the uncertainty of $ACI_r$ corresponds to the 95% confidence interval. Note that the $ACI_r$ values from six LWP bins show a generally decreasing trend of $ACI_r$ with increasing LWP (Fig. 3b). Particularly, this decreasing trend is more obvious in a range of LWPs that are less than 150 $g\,m^{-2}$. The higher values of $ACI_r$ at lower LWPs indicate that the clouds are more susceptible to aerosol loading under lower liquid water availability. Given that the $ACI_r$ describes the response of $r_e$ to $N_{CCN}$ change, under low LWP conditions with more CCN entering the cloud, the smaller particles compete against each other for the limited water supply and cannot efficiently grow into larger sizes. In that case, the higher CCN loading could result in smaller $r_e$, and thus the variable range of $r_e$ is relatively broad, which is reflected by enhanced $ACI_r$. Under high LWP conditions typically associated with sufficient water supply, the newly activated cloud droplet can grow larger quickly via condensation. However, the efficacy of condensational growth decreases with enlarged particle size. The enhanced condensational growth under high LWP conditions can shift the cloud droplet population to larger sizes. Therefore, for a similar CCN perturbation, the variable range of $r_e$ is narrower, which is reflected by reduced $ACI_r$.

Previous studies have focused on the aerosol-cloud interaction in stratocumulus clouds at the ARM SGP site. Based on the analysis of seven selected stratocumulus cases during the period 1998 - 2000, Feingold et al. (2003) reported the first ground-based measured $ACI_r$ values of 0.02 to 0.16 using the lidar measured aerosol extinction at a wavelength of 355 nm as the proxy for aerosol loading. A later study conducted by Feingold et al. (2006) assessed the $ACI_r$ using different aerosol measurements as CCN proxies, in three selected stratus cases during the intensive operation period in May 2003. They found that the $ACI_r$ values were unrealistic when

using $N_a$ to represent CCN loading while using the surface aerosol scattering coefficient ($\sigma_{sp}$) and aerosol extinction at an altitude of 350 m as CCN proxies yield similar $ACI_r$ values ranging from 0.14 to 0.39 (Feingold et al. 2006). A recent study conducted by Sena et al. (2016) within the SGP region showed the different methodologies in calculating $ACI_r$. In particular, different retrieval methods of $r_e$, could induce large differences. Moreover, the assessment of $ACI_r$ can be largely affected by the usage of different aerosol measurements that served as CCN proxies due to their own characteristics. Aerosol scattering and extinction coefficients are known to be relatively reliable CCN proxies since they are more sensitive to aerosols that have larger particle sizes. As for $N_a$, which represents the concentration of aerosol particles with diameters larger than 10 nm, it is likely to pick up the very small aerosols generated by new particle formation events. This proportion of aerosols is presumably hard to activate as CCN, so that would not be counted in $N_{CCN}$, especially under the 0.2% supersaturation used in this study. Hence, it is less representative to use $N_a$ to accurately represent $N_{CCN}$ without the prior knowledge of the aerosol capacity to activate as CCN. Therefore, the usage of $N_{CCN}$ in this study is favorable to yield a more straightforward assessment of $ACI_r$ since the CCN measurement directly represents the number of aerosol droplets that already activated and have the potential of further growth.

In order to better understand the aerosol particle activation process in typical continental low-level stratus clouds, the ratios between $N_{CCN}$ and $N_a$ are examined and used to represent the aerosol activating capacities in the latter part of this study. Aerosol activating capacities greatly depend on size and composition. In order to further examine the role of aerosol species in the aerosol activation process and the potential impact on $ACI_r$, the samples from the 16 selected cases are divided into two groups according to their absorptive regime, which is discussed in the following section.

### 3.3    Relationship between aerosol absorptive properties and ACI

### 3.3.1    Aerosol absorptive properties of the 16 selected cases

The measured absorptive properties of aerosols can aid in inferring the general information of different aerosol species since different types of aerosols can demonstrate different absorptive behaviors at certain wavelengths. Aerosol plumes dominated by organic

carbonaceous particles tend to represent strong absorptive capabilities in the visible spectrum but weakly absorb in near-infrared (Dubovik et al., 2002; Lewis et al., 2008) while black carbon particles (e.g., soot) absorb across the entire solar spectrum with a weak dependence on wavelength (Schuster et al., 2005; Lack and Cappa, 2010). However, when the aerosol plume is dominated by anthropogenic inorganic pollution, the absorbing ability becomes even weaker (Clark et al., 2007), partly due to sulfate chemical species (Chin et al., 2009). Therefore, the general existence of carbonaceous and pollution particles can be inferred via absorptive properties.

In this study, we adopt the classification method involving AE and the ratio of aerosol absorption coefficient to total extinction coefficient or single scattering co-albedo, ($\omega_{abs} = \sigma_{abs}/(\sigma_{abs} + \sigma_{scat})$) (Logan et al., 2013; Logan et al., 2014). This parameter is more sensitive to the capabilities of aerosol light absorption (rather than scattering) to total aerosol light extinction and therefore can better infer the aerosol composition (Logan et al., 2013). The $\omega_{abs}$ values at a wavelength of 450 nm along with the $AE_{450-700nm}$ of all the samples are shown in Fig. 4. A $\omega_{abs}$ value of 0.07 is used as a demarcation line of aerosols that are weakly and strongly absorbing. This value was determined using a frequency analysis performed at four AERONET sites that are dominated by single aerosol modes (Logan et al., 2013). Of the 16 cases, six cases are dominated by strongly absorbing aerosols, six cases are dominated by weakly absorbing aerosols, and four cases have samples which broadly scatter across the $\omega_{abs}$ domain, which denotes a mixture of different absorbing aerosol species.

Within the 693 selected samples, 360 data points are classified in the weakly absorptive aerosol regime, while the remaining data points are in the strongly absorptive aerosol regime. It is interesting to note that the majority of the winter cases are dominated by weakly absorbing aerosols while most of the spring cases exhibit a strongly absorbing aerosol dominance, which suggests that the aerosol plumes over the SGP site also have a seasonal dependence. In spring, owing to the upper-level ridge centered over the western Atlantic, the SGP is located at the northwestern edge of the sub-tropical high. Under this synoptic pattern, the SGP is under the influence of relatively frequent southerly transport of the airmasses from Central America, which is characterized by strongly absorbing carbonaceous aerosols produced from biomass

burning, as well as the moisture transported from the Gulf of Mexico. During the winter, the SGP site experiences the transported airmasses from higher latitudes with less intrusion of airmasses from the south (Andrews et al., 2011; Parworth et al., 2015; Logan et al., 2018).

### 3.3.2   Aerosol and cloud properties under different absorptive regimes

Figures 5a-5c show the PDFs of total $N_a$, $N_{CCN}$, and AE for the two absorptive regimes classified by $\omega_{abs}$. The distributions of $N_a$ from the two absorptive regimes are comparable to one another. The mean $N_{CCN}$ for the weakly absorptive regime (559 $cm^{-3}$) is larger than that from the strongly absorptive regime (384 $cm^{-3}$), and the occurrence of high $N_{CCN}$ values (larger than 1000 $cm^{-3}$) is also higher in the weakly absorptive regime. This suggests different responses of CCN concentration to aerosols that have similar magnitudes but different absorptive properties. The AE distributions suggest dominant fine mode aerosol contributions for both regimes. As for the cloud microphysical property distributions, cloud samples between the two regimes exhibit different characteristics (Fig. 5d-5f). The numbers above the bars in LWP distribution (Fig. 5d) for the two absorptive regimes denote the number of data points which will be used in the analysis with binned LWP in the later sections. Cloud LWPs and $r_e$ values under the strongly absorptive regime have larger values, which contrasts with those under the weakly absorptive regime. On average, the weakly absorbing regime has higher $N_d$ and smaller $r_e$ (374 $cm^{-3}$ and 6.9 µm, respectively) compared to the strongly absorbing regime (214 $cm^{-3}$ and 8.2 µm). Note that the LWPs under the strongly absorptive regime are generally higher those under the weakly absorptive regime. This LWP difference might be associated with the seasonality of airmass transport over the SGP as discussed in section 3.3.1. Although the seasonality of aerosol distribution and LWP have similar trends, no clear causality has been found between them. Thus, the question behind these results is whether the differences in cloud microphysical properties between the two regimes are due to the difference in LWP. As previously stated by Dong et al. (2015), cloud droplets generally grow larger at higher LWPs, which eventually leads to lower droplet number concentration.

### 3.3.3   Relationship of aerosol activating as CCN under different absorptive regimes

The measured $N_a$ and $N_{CCN}$ under the strongly and weakly absorbing aerosol regimes are plotted in Fig. 6. Note that $N_a$ samples from both regimes cover a broad range of values from

200-3500 $cm^{-3}$, suggesting a wide variety of aerosol loading conditions. These highly overlapping distributions allow a quantitative comparison between the ratios of $N_{CCN}$ to $N_a$. For a broad range of $N_a$, especially 200-500 $cm^{-3}$ and 1100-3500 $cm^{-3}$, the majority (~74%) of sample points from the strongly absorbing regime are located below the samples from the weakly absorbing regime. The linear regressions (95% confidence level) between $N_{CCN}$ and $N_a$ for two regimes demonstrate the sensitivity of $CCN_{0.2\%SS}$ to total aerosol loading. Note that the slope derived from the weak regime is slightly steeper than the strong regime, indicating that the $N_{CCN}$ values in the weakly absorptive regime increase faster than in the strongly absorptive regime with the same amount of aerosol increment. On average, 54% of weakly absorbing aerosols can effectively activate as CCN compared to 45% of the strongly absorbing aerosols. Note that those ratios are computed for an observed supersaturation level of 0.2%. The fraction of aerosols that can activate as CCN increases with an increase in supersaturation level, under the same aerosol size and composition condition (Dusek et al., 2006). A sensitivity test of how the aerosol activation ratio varies with different supersaturation levels is done by first interpolating the $N_{CCN}$ from 0.2% to 1.15% and then calculating the $N_{CCN}/N_a$. As a result, the ratios of $N_{CCN}/N_a$ for the weakly absorptive regime range from 0.54 to 0.38, while the ratios for the strongly absorptive regime range from 0.45 to 0.25. Considering that a supersaturation of 1.15% in continental boundary layer stratus is nearly impossible to reach, the supersaturation level of 0.2% used in this study, which represents the most typical condition for continental low-level stratus, yields reasonable results.

Although it is generally considered that the role of aerosol particle size distribution is more important than the chemical component in terms of becoming CCN (Seinfeld and Pandis, 2006; Dusek et al., 2006), many studies have found that aerosol chemical composition can also have a non-negligible impact on the aerosol activating ability under polluted and low supersaturation conditions (Rose et al., 2011; Che et al., 2016). According to Kohler theory, the critical level of supersaturation for aerosol activation depends on the aerosol solubility, which decreases with increasing soluble particle number concentration. Hence, the role of aerosol chemical composition is more important at lower supersaturation and diminishes with increasing supersaturation levels (Zhang et al., 2012).

As discussed in section 3.3.1, both weakly and strongly absorptive regimes are linked to aerosol plumes that are dominated by pollution and carbonaceous aerosols, respectively. Therefore, the difference in the ability of aerosol activation between the two regimes can be explained by the different hygroscopicity factors of the particle types. For example, anthropogenic pollution is associated with inorganic particles that are highly hygroscopic and have great ability in taking up water (Hersey et al., 2009; Massling et al., 2009; Liu et al., 2014), while carbonaceous species (e.g., black and organic carbon) exhibit varying degrees of hygroscopicity with species dominated by hydrophobic soot and black carbon being the least hygroscopic (Shinozuka et al., 2009; Rose et al., 2010). Thus, for the given amount of aerosol loading, aerosols in the weakly absorptive regime can better attract water vapor molecules and result in more aerosol particles activating as CCN.

As shown in Fig. 6, for three $N_a$ ranges (200 - 500; 500 - 1100 and 1100 - 3500 $cm^{-3}$), the strongly absorbing aerosols show different relationships compared to weakly absorbing aerosols. The mean $N_{CCN}/N_a$ values for those three $N_a$ ranges for weakly absorptive regimes are 0.77, 0.58, and 0.42, respectively, while the mean $N_{CCN}/N_a$ values for the strongly absorptive regimes are 0.35, 0.51, and 0.32, respectively. This phenomenon is due to the mixed effect of aerosol composition (inferred by absorbing ability), aerosol size, and water availability on the aerosol activation. In the 200 - 500 $cm^{-3}$ $N_a$ range, where the samples from the two absorptive regimes are most separated. The mean values of LWP (158 $g\,m^{-2}$/162 $g\,m^{-2}$ for weakly/strongly absorptive regimes) indicate relatively sufficient water availability with less aerosol concentration. In addition, the weakly absorbing aerosol sizes are larger (AE = 1.59) than the strongly absorbing aerosol (AE = 1.73). It is known that larger aerosol particles easily activate under the same composition (Dusek et al., 2006), considering the weakly absorbing aerosols are more hydrophilic, and thus the largest activation ratio difference among these three ranges are to be expected. The samples in the 500 - 1100 $cm^{-3}$ $N_a$ range, have AE values near 1.40 (1.53) for strongly (weakly) absorbing aerosols. The LWP for the strongly absorptive regime is 167 $g\,m^{-2}$ and 138 $g\,m^{-2}$ in the weakly absorptive regime. Hence, the combined effect of larger particles and more water in the strongly absorptive regime leads to a $N_{CCN}/N_a$ ratio close to the $N_{CCN}/N_a$ ratio in the weakly absorptive regime. The samples in the 1100 -

$3500 \text{ cm}^{-3}$ $N_a$ range exhibit smaller (AE = 1.67/1.57 for weakly/strongly absorptive regimes) aerosol particle size and less water availability (LWP = 95 $\text{gm}^{-2}$/127 $\text{gm}^{-2}$ for weakly/strongly absorptive regimes), which results in the lowest activation ratio ($N_{CCN}/N_a$ = 0.42/0.32 for weakly/strongly absorptive regimes) among the three ranges for both regimes.

Due to the lack of detailed chemical observations for all the cloud sample periods, as well as the uncertainties among aerosol optical and microphysical properties induced by aerosol transformation processes such as aging and mixing (Wang et al., 2010; Wang et al., 2018b), the bulk activation ratios revealed from this study cannot be significantly distinguished from each other. However, the effect of different aerosol species inferred by the absorptive properties with respect to aerosol activation is evident, especially at the 0.2% supersaturation level. Furthermore, in the following section, the values of $N_{CCN}/N_a$ and AE are sorted by LWP for the two absorptive regimes in order to rule out the influence of LWP and AE on aerosol activation.

### 3.3.4 LWP dependence of aerosol and CCN activation under different absorptive regimes

In order to better understand the role of aerosol activation ability in the microphysical process from aerosol to CCN and then to cloud droplet, comparisons must be considered under similar available moisture conditions due to the discrepancy of LWP between the two regimes. Accordingly, the sorted $N_a$ values by stratified LWP are presented in Fig. 7a, along with the activation ratios of $N_{CCN}/N_a$, which are denoted by solid lines. For a range of LWPs from 20-300 $\text{g m}^{-2}$, the ratios of $N_{CCN}/N_a$ under both regimes increase slightly with increased LWP. In addition, all binned $N_{CCN}/N_a$ values from the weakly absorptive regime (ranging from 0.4 to 0.6) are higher than those from the strongly absorptive regime (ranging from 0.3 to 0.5). A student's t-test is performed to test the ratio difference in each LWP bin at the 95% significance level. The results indicate the ratio differences between two absorptive regimes are statistically significant.

Taking the variation of $N_{CCN}$ into account, the activation ratios of $N_a$ to $N_{CCN}$ under low LWP conditions (<50 $\text{g m}^{-2}$) in both regimes could be simply due to the linear combination of high aerosol concentration and insufficient moisture supply, such that aerosols are competing against each other, thus resulting in a low activation ratio. However, as LWP increases, the

activation ratios tend to increase as well, especially at LWP values higher than 100 g m$^{-2}$. In fact, the values of $N_a$ in both regimes are relatively small with little variation for LWP > 100 g m$^{-2}$, while the $N_{CCN}/N_a$ ratio demonstrates a more noticeable increasing trend in the weakly absorptive regime. Despite higher aerosol loading in the strongly absorptive regime at large LWPs, there are still more weakly absorbing aerosols being activated, which corresponds to greater water uptake ability. Moreover, in every LWP bin, the AE value for the weakly absorptive regime is either higher than or very close to the AE value for the strongly absorptive regime (figure not shown). Even with relatively smaller particle sizes, under similar water availability, the weakly absorbing aerosol can better activate as CCN. In conclusion, a significant impact of aerosol composition on aerosol activation capacity, which is inferred by the aerosol absorbing capability, does exist.

As for the process from CCN to cloud droplet, a similar assessment is presented in Fig. 7b, which illustrates the $N_{CCN}$ values and the ratios of $N_d/N_{CCN}$ in relation to LWP. The ratios of $N_d/N_{CCN}$ in the weakly absorptive regime range from 0.58 to 0.86 with a mean value of 0.69, and highly fluctuates with LWP. In contrast, the $N_d/N_{CCN}$ in the strongly absorptive regime show lower values and less variability (from 0.47 to 0.64) with a mean value of 0.54. It is interesting to note that the variation of $N_d/N_{CCN}$ in the strongly absorptive regime mimics the variation in $N_{CCN}$ with LWP, indicating a relatively lower aerosol to CCN activating capacity. Therefore, the $N_d/N_{CCN}$ shows no significant dependence on LWP, which is consistent with previous studies, which suggest the response of $N_d$ to the change in $N_{CCN}$ has no fundamental relationship with LWP (e.g., McComiskey et al., 2009). In addition, the sensitivity and uncertainty of $N_d$ are examined in order to estimate the impact of $N_d$ uncertainty on the assessment of $N_d/N_{CCN}$. To assess the contributions of different input parameter uncertainties to $N_d$ retrieval, every input parameter was perturbed by its own uncertainty with other parameters held fixed. The results are as follows: (a) an increase (decrease) of LWP by 20 gm$^{-2}$ leads to 27.9% (27.6%) change in $N_d$ while an increase (decrease) $\sigma_x$ by 0.15 leads to a 50.8% (23.9%) change in $N_d$; (b) an increase (decrease) cloud thickness by 0.15 leads to a 14.5% (23.2%) change in $N_d$; and (c) an increase (decrease) in $r_e$ by 10% leads to 14.5% (23.2%) change in $N_d$. The percentage changes in $N_d$ due to different input uncertainties range from

14.5% to 50.8%, with the majority falling between 20% and 30%. Note that the largest uncertainty of $N_d$ happens when increasing $\sigma_x$ by 0.15. However, when considering that continental stratocumulus generally contains smaller droplets, one might expect their distribution width to be smaller than 0.38 (Dong et al., 1997). Therefore, the overall uncertainty of 25% compared to the aircraft in situ measurement should be a reasonable estimation. In this case, the mean ratio of $N_d/N_{CCN}$ for the weakly absorptive aerosol regime range from 52% to 86%, while the mean ratio of $N_d/N_{CCN}$ for the strongly absorptive aerosol regime range from 41% to 67%.

The overall differences in CCN conversion fractions are likely a result of the combined effects of meteorological factors and aerosol heating effect on the cloud environment. To examine the meteorological influence on CCN conversion, the LTS parameter is used to investigate the difference in the large-scale thermodynamic condition. By sorting the LTS by LWP for the two absorptive regimes, the LWP dependence on LTS can be ruled out, which can provide a better understanding of the potential role of LTS in cloud droplet development. For each given LWP bin, the weakly absorptive regime has higher LTS values than the strongly absorptive regime (figure not shown). The LTS is largely impacted by the potential temperature difference throughout the mixed layer. If a strong capping boundary layer temperature inversion is present, it will result in high LTS values and, in turn, a well-mixed boundary layer (Wood et al., 2006). Such results indicate that even under similar available moisture conditions, the more sufficient turbulence can transport the below-cloud moisture as well as the CCN that activated from weakly absorbing aerosols into the cloud more efficiently, contributing to a higher ratio of $N_d/N_{CCN}$ in the weakly absorptive regime. However, the LTS emphasizes a general thermodynamic condition in the lower troposphere with a wider domain as compared to the single-point measurement.

In addition, the vertical velocity in pressure coordinate (omega) values at the 925 hPa level, which represent the large-scale forcing on the vertical motion between surface and cloud-layer, are also sorted by LWP for the two absorptive regimes in order to check the potential influence of the environmental dynamic state (figure not shown). However, the omega values for both absorptive regimes share the same mean value of 0.031 Pa/s and show no dependence

on LWP, indicate that the large-scale environments over the SGP are generally dominated by sinking motion. The synoptic patterns of composite geopotential height for the two absorptive regimes show that the ARM SGP site is located ahead of the 700 hPa ridge and is located within the surface high pressure. The meteorological pattern is favorable for the generation of downward motion at the lower troposphere, and the sinking motion induces relatively stable environments in the lower troposphere, which is consistent with the LTS measurements. Considering the fact that the omega value is obtained from a relatively larger domain surrounding the SGP, it is difficult to reflect the true cloud-scale dynamics, especially the vertical velocity or turbulence strength at the cloud base. Therefore, the influence of cloud-scale dynamics, presumably cloud-base updraft, is not negligible since the sensitivity of cloud droplet to aerosol loading is enhanced with increasing updraft velocity as reported in previous studies (e.g., Feingold et al., 2003; McComiskey et al., 2009).

Furthermore, the heating effect of light-absorbing aerosols on the cloud environment cannot be neglected. Strongly light-absorbing aerosols can absorb solar radiation and heat the in-cloud atmosphere by emission, which results in the reduction of relative humidity (or supersaturation) in the cloud layer (Bond et al., 2013; Wang et al., 2013). This effect is evident by the observation as the values of in-cloud relative humidity in the strongly absorptive regime are slightly lower than those in the weakly absorptive regime. Additionally, this aerosol heating effect disrupts the boundary layer temperature structure by enhanced warming aloft, and consequently, inhibits the vertical transport of sensible and latent heat between surface and cloud layer. The impacts of light-absorbing aerosol on cloud-scale thermodynamics and dynamics state might eventually dampen the conversion process from CCN to cloud droplet. Unfortunately, due to the lack of measurement of cloud-base vertical velocity throughout the studying period, this competing effect of cloud thermodynamic and dynamic cannot be fully untangled from the aerosol effect given the currently available dataset. The differences in $N_d/N_{CCN}$ between the two regimes might be affected by the combined effects of LTS, updraft velocity, and aerosol heating effect on the cloud environment.

### 3.3.5 $r_e$ and $N_d$ dependence of LWP under different absorptive regimes

In the previous section, we examined the activation ratios of aerosol to CCN and then from CCN to cloud droplet between the two regimes as well as their dependences on LWP, which eventually led to the cloud droplet variation for a given LWP range. Figures 7c-7d demonstrate that $r_e$ increases while $N_d$ decreases with increased LWP up to roughly 150 g m$^{-2}$ in both regimes. Note that as LWPs greater than 150 g m$^{-2}$, $N_d$ values in both regimes show less variation with LWP while $r_e$ values in the strongly absorptive regime also show little variation, which implies limited growth even with increasing water availability. However, the $r_e$ values in the weakly absorptive regime increase from 7.8 to 8.8 μm, which suggests that under a given number concentration, the cloud droplet can grow by continuing to collect moisture. As shown in each LWP bin, the $r_e$ values in the weakly absorptive regime are smaller than those in the strongly absorptive regime, while the $N_d$ values in the strongly absorptive regime are much lower than those in the weakly absorptive regime.

The combination of cloud thermodynamic, dynamic, and light-absorbing aerosol heating effects impacts the conversion process from CCN to cloud droplet. Under a given moisture availability, a higher number of CCN in the weakly absorptive regime can be converted to cloud droplets. This results in higher number concentrations of smaller cloud droplets, while the dampened CCN conversion process in the strongly absorptive regime leads to fewer and larger cloud droplets at a fixed LWP.

### 3.3.6  Aerosol-cloud-interaction under different absorptive regimes

To examine the sensitivity of clouds to both weakly and strongly absorbing aerosol loading, the relationships between cloud $r_e$ and $N_{CCN}$ are shown in Fig. 8. Two LWP ranges (0-50 g m$^{-2}$ and 200-250 g m$^{-2}$) are selected in order to better represent ACI$_r$ at low and high LWP conditions. For the examination of $r_e$ as a function of $N_{CCN}$ (Fig. 8a for low LWP range), the ACI$_r$ values in the weakly absorptive regime are higher than those in the strongly absorptive regime. This suggests that the cloud droplets are more sensitive to weakly absorbing aerosols than to strongly absorbing aerosols in clouds with low LWPs. In other words, if there is some increment in aerosol particles, clouds influenced by weakly absorbing aerosols will respond to this increment more effectively and decrease faster in droplet sizes relatively. Under high LWP conditions (Fig. 8b), the ACI$_r$ values are lower and show less difference between the two

regimes, which is in agreement with previous discussions on the sensitivity of cloud microphysical properties to aerosol loading.

Based on the sensitivity study, the 10% change of cloud LWP and downward SW at the surface would result in the 10% uncertainty in $r_e$ retrieval (Dong et al., 1997). When compared with aircraft in situ measurements, the differences between retrievals and in situ measurements are around 10% (Dong et al., 1998, 2002). In order to assess the impact of $r_e$ uncertainty on $ACI_r$, we use the Monte Carlo method to propagate the $r_e$ uncertainty on $ACI_r$ by the following procedure. The $r_e$ value for each data point is randomly perturbed by ±10%, and thus the corresponding $ACI_r$ can be re-calculated based on the perturbed set of $r_e$. After 100,000 iterations, we obtain a distribution of $ACI_r$s. The uncertainty of $ACI_r$ is given by one standard deviation of those 100,000 values of $ACI_r$s, since the distribution of $ACI_r$s follows a normal distribution with a narrow peak, this uncertainty value represents the uncertainty in the computed $ACI_r$ due to errors in the $r_e$ retrieval. The uncertainties of $ACI_r$ for the two absorptive regimes are denoted as the dashed line in Fig. 8. In the lower LWP range (Fig. 8a), the $ACI_r$ uncertainty is 0.020 (0.030) for the weakly (strongly) regime, account for the uncertainties, the difference in $ACI_r$ between the two absorptive regimes is preserved. In the higher LWP range (Fig. 8b), the $ACI_r$ uncertainty is 0.044 (0.023) for the weakly (strongly) regime, which is non-negligible. Taking the uncertainties of $ACI_r$ into account, the $ACI_r$ in the two absorptive regimes cannot be well separated, owing to the enhanced condensational growth process accompanied by higher LWP and the diminished cloud response to aerosols associated with different $\omega_{abs}$ values. In general, the 10% uncertainty in $r_e$ retrieval contributes to 0.02 ~ 0.04 in $ACI_r$ uncertainties.

Note that the LTS values from the weakly absorptive regime (22.91K and 19.78K) are higher than those from the strongly absorptive regime (21.72K and 17.83K) for the selected two LWP bins. As discussed in the previous section, owing to the stronger temperature inversion indicated by the higher LTS values, low clouds are more closely connected to weakly absorbing aerosols and moisture below cloud by sufficient turbulence. In order to quantify the impact of LTS on $ACI_r$, we adapted the criteria described in Grysperdt et al. (2016) that the LTS value of 18 K denotes the demarcation line between high and low LTS regimes, and

constrain the $ACI_r$ for the two regimes by their LTS values accordingly. Owing to the highly limited sample points that fall into the low LTS category, the $ACI_r$ can only be constrained in the high LTS condition. For the 0 - 50 $g\,m^{-2}$ LWP range, the $ACI_r$ for the weakly absorptive regime increases from 0.26 to 0.31, and the $ACI_r$ for the strongly absorptive regime increases from 0.21 to 0.24. The enhancement effect of LTS on the $ACI_r$ is noticeable, which in accordance with the previous discussion that high LTS environment is associated with (a) sufficient turbulence in the boundary layer and (b) a closer connection between the surface and cloud layer, which enhances the cloud microphysical responses to the CCN. The result is consistent with the previous study by Kim et al. (2008) who found that $ACI_r$ is enhanced under adiabatic cloud conditions and higher LTS values are associated with higher cloud adiabaticity. Note that though $ACI_r$s are increased for both regimes, the difference between the regimes becomes larger (from 0.05 to 0.07) because low-level stratus clouds are more susceptible to weakly absorptive aerosol. Furthermore, the enhancement of $ACI_r$ in the high LTS environment is more evident in the weakly absorptive regime. In the case of the $200 - 250\ gm^{-2}$ LWP range, the LTS effects on $ACI_r$ are less significant compared to the lower LWP range. No significant change in weakly absorptive regime is evident and the $ACI_r$ in the strongly absorptive regime decreased from 0.12 to 0.10, partly owing to the enhanced condensational growth process accompanied by higher LWP, and thus inhibits the impact of LTS on $ACI_r$. Overall, $ACI_r$s are enhanced under high LTS conditions, but the difference between the two regimes indicates that both $\omega_{abs}$ and LTS can be the impact factor of the $ACI_r$, but they are not necessarily having causality between them.

Furthermore, with the presence of strongly light-absorbing aerosols, the cloud layer heating induced by the aerosol absorptive effect can result in the reduction of in-cloud supersaturation and leads to the damping of cloud microphysical sensitivity to strongly absorbing aerosols. A previous modeling study conducted at the ARM SGP site by Lin et al. (2016) estimated the shortwave heating rates in cloud layers by contrasting the simulations with and without light-absorbing aerosols. The inclusion of light-absorbing aerosols was represented by an internal aerosol mixture with a mass combination of 95% ammonium sulfate and 5% black carbon. The SSA of this mixture is calculated to be roughly 0.9, as documented

in the previous study of Wang et al. (2014). The different values of SSA used in their study (0.9 for light-absorbing and 1.0 for non-absorbing) are comparable to this study (0.89 for strongly-absorbing and 0.97 for weakly-absorbing). The induced increments in cloud-layer shortwave heating rates have a maximum value of 3 K/day, compared to the simulation without light-absorbing aerosols. Note that the aerosol number concentration in Lin et al. (2016) was set to 2800 $cm^{-3}$. To get a simple comparison with the aerosol number concentration in this study, one might expect the light-absorbing aerosols induced cloud-layer shortwave heating rates can have a similar maximum increment, and the general increment should be about 1 K/day which is non-negligible. The absorption of solar radiation by light-absorbing aerosols warm the cloud layer as well as the boundary layer below it, which in turn, stabilizes the lower troposphere and results in reduced cloud susceptibility. In general, the results indicate that the $ACI_r$ can be counteracted by the light-absorbing aerosol heating effect and be enhanced under a thermodynamic environment of high static stability, especially under lower LWP conditions.

### 3.4 Cloud shortwave radiative effects under different absorptive regimes

Aerosols with different absorptive properties can alter the ability of clouds to reflect incoming shortwave radiation. Accordingly, cloud radiative effects on shortwave radiation for the two absorptive regimes are investigated. Both cloudy and clear-sky downwelling shortwave fluxes for samples in the weakly absorptive regimes are generally higher than those in the strongly absorptive regime (not shown in here), largely owing to the discrepancies in solar zenith angle, seasonal variation of insolation, and surface albedo. Therefore, to ensure the comparison is under minimum influence of non-cloud factors, the shortwave relative Cloud Radiative Effects (rCREs) are introduced and their dependencies on LWP between the two regimes are examined. With all else being equal, as shown in Fig. 9, rCREs in both regimes noticeably increase with LWP, especially for LWPs less than $150\,g\,m^{-2}$. Using fixed LWP, rCREs in the weakly absorptive regime are always higher than those in the strongly absorptive regime, because the greater activating ability of the weakly absorbing aerosols leads to higher $N_d$ and smaller $r_e$ as opposed to the strongly absorbing aerosols. Thus, clouds with a larger amount of small cloud droplets contribute more to the extinction of incident solar radiation. The difference in mean rCRE between the two regimes is small but non-negligible (~0.04).

Quantitatively speaking, taking the climatological downwelling solar flux of the winter season ($\sim$150 W m$^{-2}$, Dong et al., 2006) as an example, the extinction of incident solar radiation by clouds that develop from weakly absorbing aerosols is 6.0 W m$^{-2}$ more than those by clouds from strongly absorbing aerosols. From independent radiative measurements, the phenomenon that clouds are more susceptible to weakly absorbing aerosols is further evident.

## 4    Conclusions

A total of 16 non-precipitating overcast low-level stratiform cloud cases under daytime coupled boundary layer conditions were selected in order to investigate the sensitivity of cloud microphysical properties to aerosol physicochemical properties. The Ångström exponent and fine mode fraction distributions indicate that the aerosol plumes that advected to the SGP site during all the selected cases were dominated by fine mode particles, while the variation in aerosol single scattering albedo suggests different characteristics of optical properties among the aerosol plumes. In terms of the sensitivity of cloud droplets to aerosol number concentration, the values of ACI$_r$ range from 0.09 to 0.24 with a mean of $0.145 \pm 0.05$, which supports the finding of previous studies using ground-based measurements. The magnitude of ACI$_r$ shows a decreasing trend with increasing LWP, partly owing to the enhanced condensational growth process accompanied by higher LWP. Clouds that develop under lower LWP conditions are more susceptible to aerosol loading, owing to the enhanced competition between small cloud droplets to grow larger with limited water supply.

The analysis of the $N_{CCN}$/Na ratio under the two regimes further demonstrates that weakly absorbing aerosols have statistically significant higher activation ratios (mean ratio of 0.54) than the strongly absorbing aerosols (mean ratio of 0.45). The fraction of weakly absorbing aerosols that activate as CCN shows a noticeable increase with increased LWP, while the activation ratios for strongly absorbing aerosols tend to slightly increase with LWP under comparable aerosol loading conditions. This is likely related to the hygroscopicity associated with the aerosol species. For example, weakly absorbing aerosols are typically dominated by pollution aerosols that have greater water uptake ability, while strongly absorbing aerosols are generally hydrophobic, such as freshly emitted black and organic carbon.

The ratios of $N_d/N_{CCN}$ in the weakly absorptive regime (mean ratio of 0.67) are higher than for the strongly absorptive regime (mean ratio of 0.54). This is due to the higher LTS environment for the weakly absorptive regime, enhancing the connection between cloud and the below-cloud moisture and CCN. In addition, the cloud layer heating effect induced by the strongly light-absorbing aerosols results in the reduction of in-cloud supersaturation and leads to the damping of the CCN conversion process for the strongly absorptive regime. As a result, cloud droplets that form from weakly absorbing aerosols tend to have smaller sizes and higher concentrations than cloud droplets forming from strongly absorbing aerosols. Furthermore, the cloud droplets in the weakly absorptive regime exhibit a greater growing ability, as given by larger $r_e$ values that increase with LWP under similar $N_d$. The differences in cloud droplet development between the two regimes is a likely result of the combination of thermodynamics, dynamics, and aerosol heating effects.

Under low LWP conditions, the measured $ACI_r$ values in the weakly absorptive regime are relatively higher, indicating that clouds have greater microphysical responses to aerosols in weakly absorptive regime than in strongly absorptive regime. The favorable LTS condition in the weakly absorptive regime enhanced the cloud susceptibility. The cloud-layer heating effect of light-absorbing aerosol reduced the $ACI_r$ in the strongly absorptive regime. The observed $ACI_r$ is enhanced after being constrained by high LTS, particularly under lower LWP conditions. Under higher LWP conditions, the enhanced condensational growth process diminishes the LTS impact on $ACI_r$, and the damping of $ACI_r$ is more evident, which is consistent with the results from all the cases. In general, the 10% uncertainty in $r_e$ retrieval contribute to $ACI_r$ uncertainties range from 0.02 to 0.04 for the two absorptive regimes, with the $ACI_r$ difference between the two absorptive regimes is still well-preserved. As a result, clouds that develop from weakly absorbing aerosols exhibit a stronger shortwave cloud radiative effect than clouds originating from strongly absorbing aerosols. Additional future work will focus on investigating in detail the composition of different aerosol plumes with respect to their physicochemical properties. The aerosol-cloud-interaction processes under the influence of different aerosol types associated with airmasses and the sensitivity to dynamic and thermodynamic factors will be further examined.

**Data availability.** Data used in this study can be accessed from the DOE ARM's Data Discovery at https://adc.arm.gov/discovery/ (last access: 17 July 2019).

**Author contributions.** The original idea of this study has discussed by XZ, BX, and XD. XZ performed the analyses and wrote the manuscript. XD, TL, YW, and PW participated in further scientific discussion and provided substantial comments and edits on the paper.

**Competing interests.** The authors declare that they have no conflict of interest.

**Acknowledgements.** The ground-based measurements were obtained from the Atmospheric Radiation Measurement (ARM) Program sponsored by the U.S. Department of Energy (DOE) Office of Energy Research, Office of Health and Environmental Research, and Environmental Sciences Division. The reanalysis data were obtained from the ECMWF model output which specifically provides for analysis at the ARM SGP site. The data can be downloaded from http://www.archive.arm.gov/. The researchers at the University of Arizona are supported by the NSF project under grant AGS-1700728, Dr. Timothy Logan is supported by National Science Foundation Collaborative Research under award number AGS-1700796 at Texas A&M University, and Dr. Yuan Wang at California Institute of Technology is supported by AGS-1700727. The two anonymous reviewers and the co-editor are acknowledged for constructive comments and suggestions, which helped to improve the manuscript.

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

**Table 1.** Dates and time periods of selected low-level stratus cloud cases and their airmass source.[a]

| Date | Start Time (UTC) | End Time (UTC) | Airmass Source | Number of Data Points |
|---|---|---|---|---|
| 4 Jan 2007 | 15:00 | 22:30 | S | 58 |
| 5 Jan 2007 | 14:00 | 18:10 | S | 40 |
| 13 Feb 2007 | 17:00 | 22:30 | N | 60 |
| 26 Apr 2007 | 14:00 | 17:30 | NE | 31 |
| 21 Nov 2007 | 13:20 | 18:15 | N | 24 |
| 14 Feb 2009 | 15:15 | 17:35 | NW | 29 |
| 12 May 2009 | 16:55 | 20:05 | SE | 37 |
| 19 Dec 2009 | 14:40 | 19:35 | NW | 58 |
| 21 Jan 2010 | 15:25 | 22:30 | N | 44 |
| 16 Mar 2010 | 15:00 | 20:00 | N | 41 |
| 29 Dec 2010 | 16:00 | 18:35 | SE | 32 |
| 26 Mar 2011 | 16:35 | 23:55 | NE | 59 |
| 13 May 2011 | 12:25 | 18:20 | N | 59 |
| 4 Feb 2012 | 16:40 | 21:10 | NE | 37 |
| 8 Feb 2012 | 14:30 | 19:45 | N | 54 |
| 10 Feb 2012 | 17:15 | 19:50 | NW | 30 |

[a]Airmass sources denote the relative directions from where the airmasses advected to the ARM-SGP site.

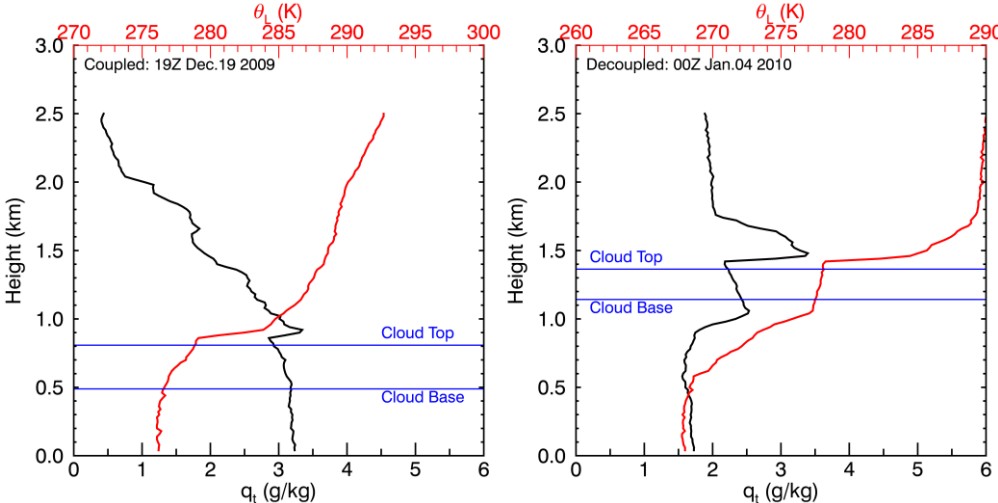

**Figure 1.** Vertical profiles of liquid water potential temperature ($\theta_L$) and total water mixing ratio ($q_t$) for coupled (a) and decoupled (b) boundary layer conditions. Blue lines denote cloud top and base heights, respectively.

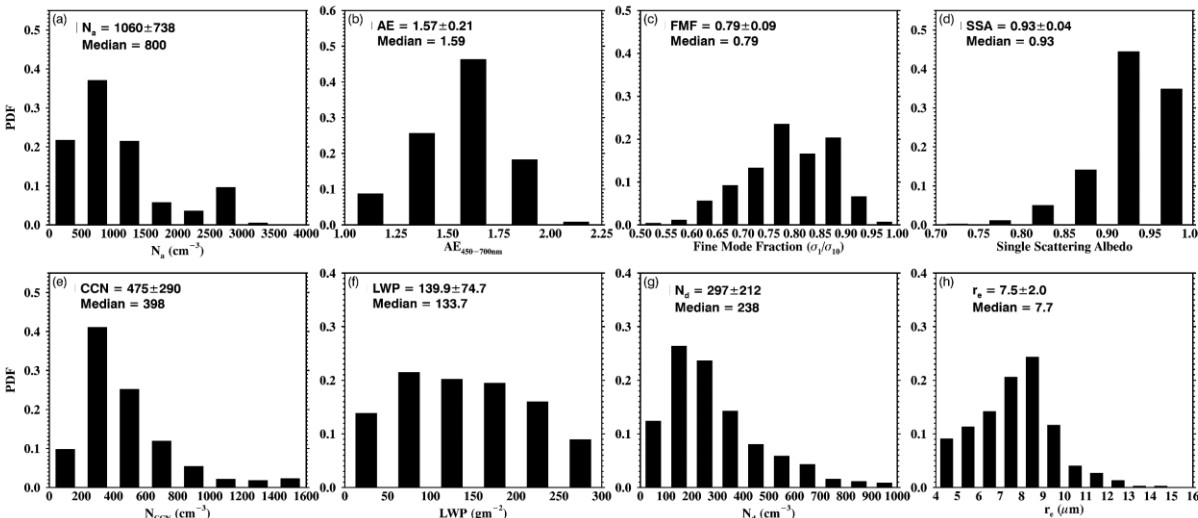

**Figure 2.** Probability distribution functions (PDFs), mean and median values of low-level stratus cloud and aerosol properties for all cases: (a) total aerosol number concentration (Na); (b) Ångström Exponent (AE) derived from nephelometer measurements; (c) fine mode fraction at 550 nm; (d) single scattering albedo at 450 nm (SSA); (e) cloud condensation nuclei number concentration ($N_{CCN}$); (f) liquid water path (LWP); (g) cloud droplet number concentration ($N_d$); (h) cloud droplet effective radius ($r_e$).

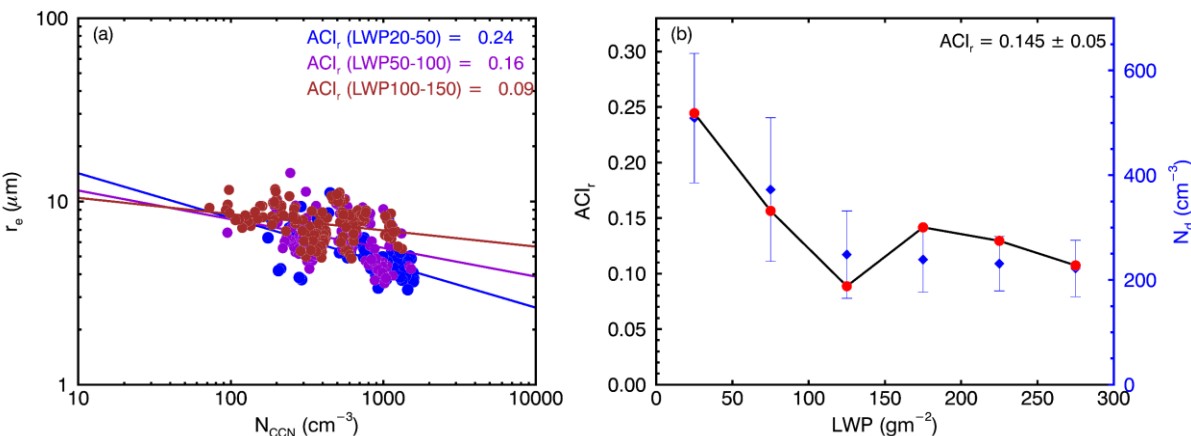

**Figure 3.** $ACI_r$ derived from (a) $r_e$ to $N_{CCN}$ in following three LWP bins: 20-50 gm$^{-2}$ (blue), 50-100 gm$^{-2}$ (purple), 100-150 gm$^{-2}$ (dark red) and (b) Relationship of $ACI_r$ (red dot, left ordinate) and $N_d$ (blue diamond, right ordinate) to binned LWP. Blue whiskers denote one standard deviation for each bin.

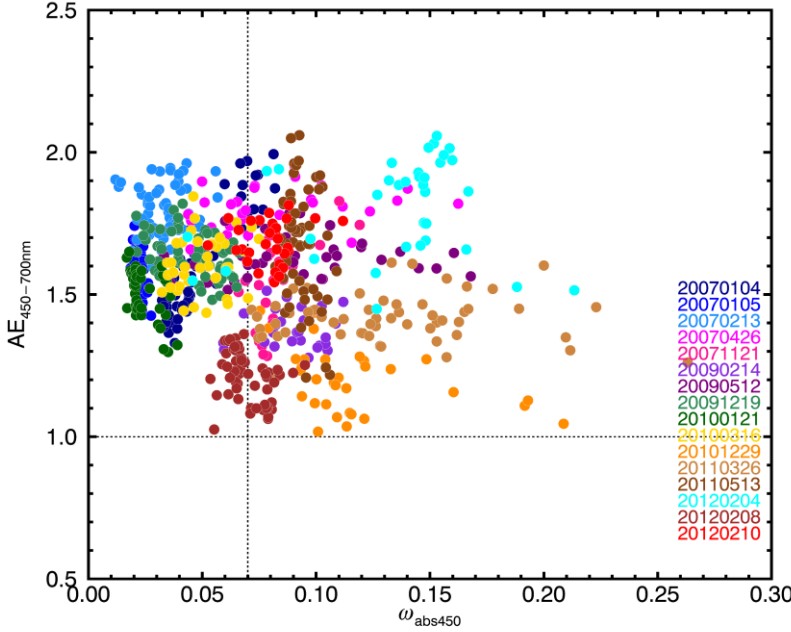

**Figure 4.** Angstrom Exponent ($AE_{450-700nm}$) and single scattering co-albedo $\omega_{abs450}$ of all samples (color coded by case). Horizontal dotted line denotes the demarcation of $AE_{450-700nm}$ = 1. Vertical dotted line denotes the demarcation of $\omega_{abs450}$ = 0.07.

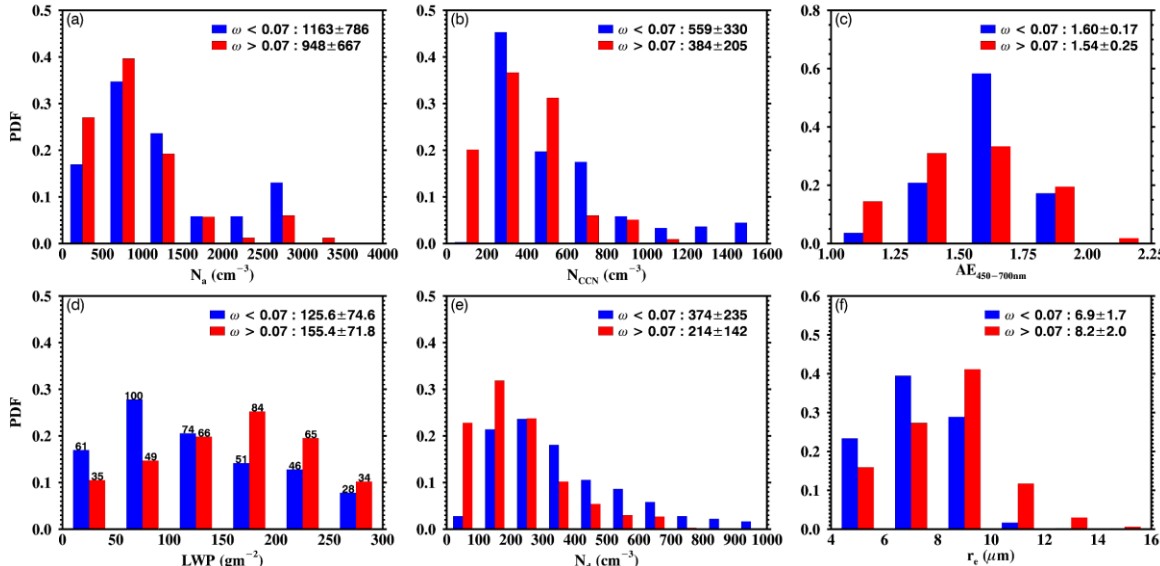

**Figure 5.** Aerosol and cloud properties under the strongly absorptive (in red) and the weakly absorptive (in blue) aerosol regimes. PDFs, mean values and standard deviations of (a) $N_a$; (b) $N_{CCN}$; (c) $AE_{450-700nm}$; (d) LWP; (e) $N_d$; (f) $r_e$.

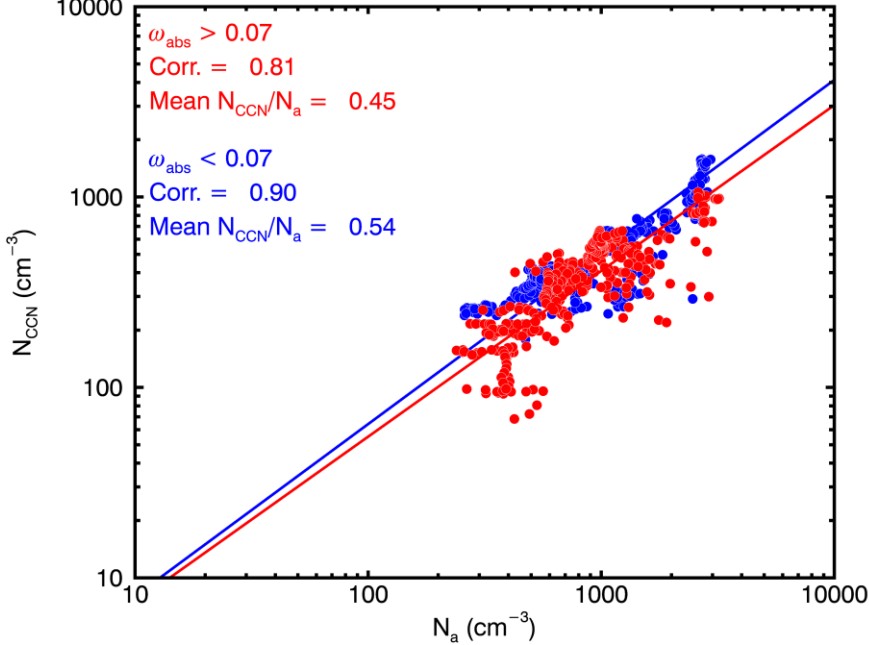

**Figure 6.** Relationship between $N_{CCN}$ and $N_a$ under the strongly absorptive aerosol regime (in red) and the weakly absorptive aerosol regime (in blue).

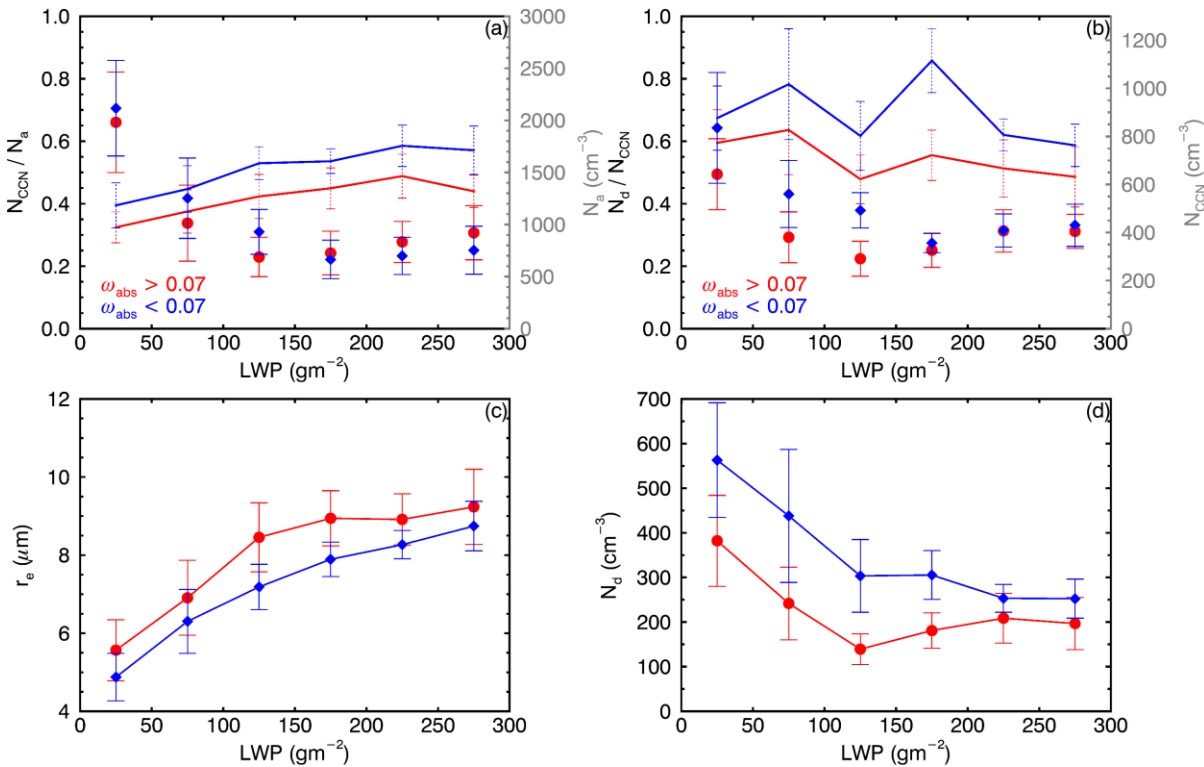

**Figure 7.** (a) $N_a$ (dot) and the ratio of $N_{CCN}$ to $N_a$ (line); (b) $N_{CCN}$ (dot) and the ratio of $N_d$ to $N_{CCN}$ (line); (c) $r_e$; and (d) $N_d$ as a function of LWP under strongly absorptive (in red) and weakly absorptive (in blue) aerosol regimes. Whiskers denote one standard deviation for each bin.

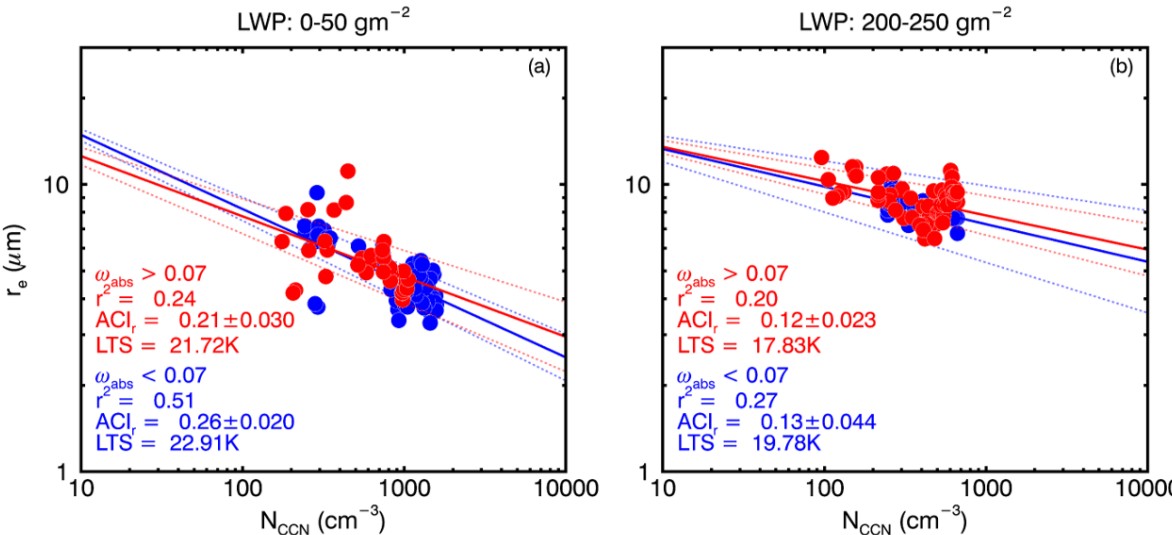

**Figure 8.** $r_e$ as a function of $N_{CCN}$ and the values of $ACI_r$ under the strongly absorptive (in red) and the weakly absorptive (in blue) aerosol regimes at two LWP bins: 0-50 g m⁻² (a) and 200-250 g m⁻² (b). Note that the dashed lines denote the uncertainties of $ACI_r$ due to 10 % error in $r_e$ retrieval.

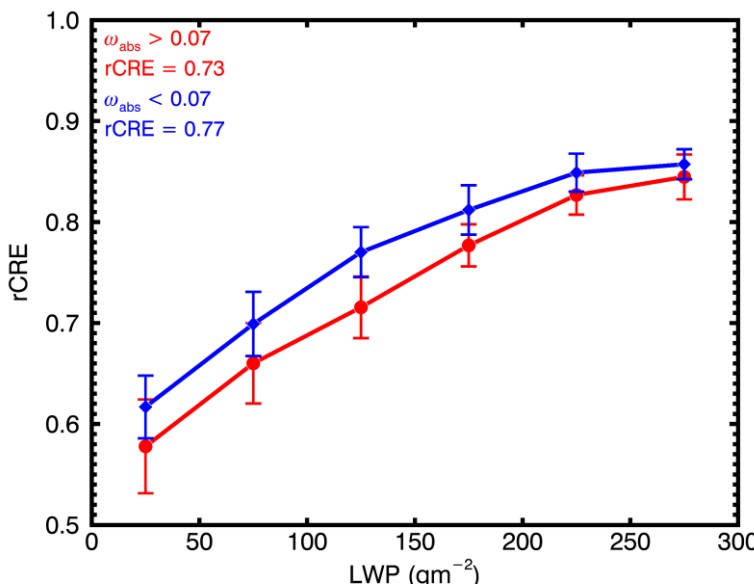

**Figure 9.** Relative Cloud Radiative Effect (rCRE) as a function of liquid water path (LWP) under the strongly absorptive (in red) and weakly absorptive (in blue) aerosol regimes. Whiskers denote one standard deviation for each bin.