# Peer review of "Investigation of Aerosol-Cloud Interactions under Different Absorptive"

_Atmospheric Chemistry and Physics, 2019_

## Referee Comment (RC1) · Anonymous Referee #1 · 10 Jul 2019

This manuscript presents a study about the effect of aerosols on cloud properties using ARM facility over the SGP. The study based their analysis on 16 low-level stratus under coupled boundary layer conditions. A comprehensive comparison is performed between weak and strong light-absorbing aerosols cases. An ACI index has been retrieved for different cases: different LWP bins and different absorptivity. The article based their results on multiple parameters to asses the impacts of aerosols on cloud microphysical, optical, and radiative properties. The manuscript is well-structured and well-written, the illustrations included are useful. I expect the paper to be of consider-

able interest to readers of ACP. Nevertheless, in my view, the following issues require particular attention before a potential submission.

General Comments:

1. The uncertainties and hypotheses are mentioned in the article but a discussion about them is needed. For the hypotheses, it is referred that $\sigma_x$ is assumed constant at 0.38 (page 5 line 23): Is there a way to estimate the impact from a variation of this value according to a sensible range? The same comment goes for the supersaturation fixed at 0.2% (page 6 line 21), what are the impacts on the results if SS = 1.15%.

The parameters are associated with uncertainties, as mentioned several times in the text: LWP (page 5 line 9), re and Nd (page 5 line 25), SW fluxes (page 7 line 17), the contamination by insect (page 4 line 27). Unfortunately, these uncertainties are not considered in the study when comparing the ACI parameters for the different regimes. What are the impacts of the uncertainties on the results? Can you estimate the impacts on ACI to ensure that the observed difference is real?

2. Two "meteorological" parameters have been considered (section 2.3): the liquid water potential temperature and the total water mixing ratio. These parameters have been used to consider if the boundary layer is well-mixed or not. I think a deeper study on the impact of meteorological parameters on ACI would increase the impact of the paper. As shown in Table 1, the cases correspond to different seasons and air-mass sources. The different ACI observed for absorptive or non-absorptive aerosols might be due to different meteorological parameters (as stated in page 11 line 16) and potentially not to the difference of aerosol optical properties. I think different regimes based on meteorology parameters (e.g., stability) should be considered to strengthen the results.

3. In the study, I do not understand if each day is taken separately to perform the analysis or if the study considers each measurements: For example, in Figure 4, we observe that some days have a large range of AE (2011/05/13), did the distribution of
AE shown in Figure 2-b consider each point from Figure 4 or the average for each day (total of 16 points)? I assume that it is each point but the text needs to make it clear. Therefore, I do not understand why the cloud lifetime needs to be more than 3 hours (page 7 line 26) if each measurement is considered independently

Specific Comment:

There is no indication of how many data points are considered in the analysis.

Abstract: A sentence about the context, and why it is important to study the aerosol cloud interaction is missing.

page 3 line 10: I suggest to remove the "co-albedo" has it can confuse a reader which is not familiar with this term, or to specify that it is 1-SSA.

Page 4 line 20: The study uses two different instruments with different spatial and temporal resolutions, how does it affect your results? Is the uncertainty from the KAZR lower?

page 5 line 4: Why has the "cloud-top height lower than 3 km" limit been chosen?

page 6 line 14: What is the initial temporal resolution?

Is there a study comparing the measurements from SGP with in-situ data to evaluate the performance of the instruments? The results are provided for cloud microphysical properties (page 5 line 25), but is there something similar for the aerosol properties , cloud boundaries, and boundary layer conditions?

page 7 line 6: 0.5 K and 0.5 g/kg: are these thresholds the same as in Dong et al. (2015)?

page 8 line 24: Can you describe the difference between FMF and AE? I am not sure to understand why the study needs the two parameters.

page 10 line 5: How is the uncertainty on ACI retrieved? Is it the 95% confidence

interval of the fit?

page 10 line 6-: The authors are comparing ACI values with previous studies. I am a bit sceptical about it: There is plenty of studies retrieving ACI values with different methods, datasets, geographical locations. ACI parameter depends on that. The authors only report ACI values which range with their study without a discussion on the differences. I think there is two different possibilities: Either you consider all the studies retrieving ACI and discuss about the potential differences or the comparison is limited to studies looking at the same region and/or same data.

page 11 lines 23-26: I do not understand the sentence, can you rephrase it?

page 11 line 26-27: Can it be quantified?

page 12 line16: The "majority", can it be quantified?

Figure 2: Can the standard deviation be displayed with the mean? Also, considering that the distributions are not Gaussian, why did you consider the mean rather the median?

Figure 7: Is there a reason why the standard deviation is not included for the ratios NCCN to Na and Nd to NCCN?

Technical corrections:

page 5 line 18 "100" should not be here.

page 8 line 17: find → fine

page 15 line 25: Fig. 10 → Fig. 8

Figure2 caption: the order to describe the figures is: a-d-b-c-. . . instead of a-b-c-d-. . .

Figure 4: The definitions of the dotted lines are missing in the caption?

---

## Referee Comment (RC2) · Anonymous Referee #2 · 12 Aug 2019

Overview: This study uses data from 16 low stratus cloud cases at the ARM SGP site in Oklahoma to determine the effect of aerosol absorption on commonly used aerosol-cloud interaction index, specifically the change in cloud drop effective radius with a change in aerosol (ACIr.) This approach is of interest as aerosol composition may impact cloud drop activation and ultimately aerosol indirect effects. The authors follow methods from existing literature very closely with the novel piece of adding information on aerosol absorption. It is understood that aerosol compositions that are more highly absorbing can be lass hygroscopic, thus affecting aerosol indirect effect so overall this

study is of interest. The data do indicate that aerosols that are less absorbing are more likely to be suitable as CCN that aerosols that are more absorbing. There are significant shortcomings however in the relationship from CCN to cloud properties as meteorological parameters that may be mediating cloud properties and may or may not be related to the aerosol absorption are not addressed. These topics as outlined below should at least be discussed and the potential implications included in the manuscript (if a further analysis of meteorology/radiative effects is not included) prior to publication. I've categorized these as minor revisions but could take some doing.

General: Temporal resolution and number of data points While the temporal resolution of some of data products used to create the data set is given (I think everything is averaged to 5-minutes but it's not explicitly noted anywhere), nowhere is it stated the number of data points used to determine the ACIr index and other correlations. The 16 cloud cases and total time that the data set covers is provided in Table 1 but does not contain these statistics, which are important for interpreting results. If I do some math it seems that there are sufficient statistics but text needs to be added to fully and clearly describe the statistics of the data set for the reader.

Choice of aerosol data used - The authors choose to use the sub-micron aerosol optical properties from the available measurements rather than the sub-10 um. What is the motivation for this choice? The total aerosol number concentration Na and CCN are used which are not restricted to the sub-micron size cut. It's not fully consistent that the Na and CCN be sorted by high and low absorbing regime according to the sub-micron absorption – there could be a relationship between size and composition/absorption. Further, the sub-micron scattering fraction is presented alongside the scattering angstrom exponent for the sub-micron aerosol only. This lacks consistency and can make interpretation of the results difficult when reading through progressive steps in the analysis. An explanation (and implications) for how the sub-micron only properties relate to the others should be given if the choice of data is not changed. It probably won't change the overall picture given that the aerosol is largely sub-micron

but why complicate the issue?

Lack of meteorological parameters or aerosol radiative effects in assessing co-variances of aerosol and cloud properties - My greatest concern with this analysis relates to the association of aerosol absorption to cloud microphysics and cloud radia-tive effect without considering meteorological or systematic seasonal influences that may be affecting the co-variance of aerosol and cloud properties. The relationship of Na to CCN for high and low absorption regimes is compelling and it does seem that the difference in composition has an effect on the number of CCN. But then the ex-amination of the relationship between CCN and drop number is presented without any discussion of controls by cloud dynamics or potential radiative effects of the absorbing aerosol on the environment of cloud dynamics.

Related is the fact that most cases occurred only during winter and spring and largely under northerly wind conditions. Also, the authors state that the high and low absorp-tion cases split largely along the same lines with higher absorption occurring in spring, however the implications of the co-variability in aerosol and cloud properties is never discussed. You note that the LWP is larger under the high aerosol absorption regime – is this causal? A seasonal effect? You also note that higher absorption occurs in Spring. This fact is not revisited and explained in the discussion after all relationships have been analyzed. These two factors could be unrelated but both driven by seasonal effects on aerosol distributions and available moisture separately. What implications would this have for the relationships you present here?

In Fig 8, why look at drop effective radius as a function of Na if you have CCN mea-surements and have already established the Na to CCN relationship dependence on absorption? I feel like the effect of size and composition on drop activation are getting conflated here and in some other places in the manuscript. Given that the absorption dependence of Na to CCN is compelling, you might to better to simplify the paper by omitting some of these plots that don't add to the message and can actually be con-fusing. On P16, the last paragraph of section 3.3.6 has a related discussion that is

Interactive
comment

confusing. The relationship of Na to clouds and CCN to clouds is considered. This doesn't make sense – the definition of CCN is the segment of the total aerosol population that will activate to form cloud drops (the statement 'clouds are more sensitive to CCN than solely aerosol particles' should be deleted.) In reality the number of cloud drops might not equal CCN, but if the measurements are good then that is due to some competing effect of cloud dynamics, available moisture, radiative effects, etc. (none of the latter are addressed here.) Other statements in the paper that follow this confusing logic are P18 L4-5 '…conversion rates of Nd/Nccn for weakly absorbing aerosols are higher than for strongly absorbing aerosols' suggests that there is some other mechanism at play like a radiative effect – or the CCN measurement is not accurate. Also P18 L13-14 '..the mechanism from CCN to cloud droplet is more straightforward than from aerosol particle to cloud droplet.' It ought to be.

Misc: The discussion at the top of P10 regarding results from other studies that calculated ACI should include the parameters and sampling used in those studies to provide some background on why values might differ. Some discussion of the results rather than a simple reporting of the numbers should be included. How the data is sorted and what dependencies are examined can have a large impact on this indexes due to the inherent sensitivity of cloud microphysics to a range of parameters.

In Fig 7 are the differences in the ratios statistically significant? Where standard deviations are included it's easier to judge what might be significant, but there is a general lack of discussion of uncertainties and statistical significant throughout. This is further complicated by the lack of information on the number of data points used in each analysis (or each bin in the binned analyses) as commented above.

Figures: All of the labels need to be much bigger – many are very difficult to read. Figure 2 caption has the sub-figured listed out of order – should be ordered alphabetically from a-f Figure 3 red and orange colors are indistinguishable Figure 6 not points above the 1:1 line is curious – this almost always exists due to measurement error – were these removed?

Specific: Page and Line P2 L23: 'influence' rather that 'interact with' (suggestion)

P2 L28: 'inferred' rather than 'identified by' (suggestion) – your explanation of the uses and limitations of inferring composition from optical properties is quite nice

P3 L6-9: may also not that measurements of absorption angstrom exponent typically carry large uncertainties

P7 L25-27: and the restriction of LWP > 20 g m-3

P7 L27: what is the reasoning behind the daytime only? Simply that the quantity is only available under sunlight conditions? Consider rewording

P8 L17: 'find' should be 'fine'

P8 L22-24: this sentence is confusing – maybe '. . .greater than 0.6 represent the dominance of fine mode aerosol in the total population and values less than 0.2 represent the dominance of coarse mode aerosols in the total population.'

P8 L25: 'dominated' should be 'dominant'

P9 L16: note that 'theoretical' values of ACIr. . .

P11 L9: don't think you can state that co-albedo provides information about composition, just more sensitive to the amount of absorption

P11 L22-25: sentence needs rewriting – may just need a 'For' at the start and to remove 'higher' at the end

P15 L19-22: how much does the composition of CCN matter for growth once it's already activated?

P15 L25: should be Fig 8

---

## Author Comment (AC1) · 22 Sep 2019

**Response to Reviewer #1**

We appreciate your time for carefully reviewing our manuscript. We would like to thank you for the constructive comments and suggestions, which encourage and help us to improve the manuscript. The manuscript has been revised accordingly. In the response below, the reviewer's comments are provided in black text and our responses are provided in blue text.

**Response:**

General responses and changes in aerosol data:
Reviewer #2 suggested that the original use of submicron aerosol optical properties data could cause the inconsistency in aerosol number concentration and optical properties. Therefore, aerosol optical properties have been changed to the measurements of sub-10μm size-cut in the revised manuscript and the data was re-sorted by the (1-SSA) values of sub-10μm aerosols into high and low absorptive regime accordingly. In general, the main results and conclusions did not change significantly given the fact that fine-mode aerosols dominated the aerosol plumes over SGP.

The Figures below show the corresponding changes after new aerosol results have been used in the revised version:

Figure 2: For the total dataset, the mean value of AE changes from 1.67 to 1.57; the mean value of SSA changes from 0.94 to 0.93, which results from the contribution of aerosols having a range of diameters from 1 – 10 microns.

[Figure]

Figure 4: As indicated in the revised Figure 2, the values of (1-SSA) generally increase and AE values generally decrease owing to the inclusion of aerosols having diameters greater than 1 micron.

[Figure]

Figure 5: The mean values changed slightly due to the new categories of absorptive regimes, but the differences in distributions and mean values between the two regimes were preserved.

[Figure]

Figure 6: The overall activation rates did not change, with more data points from the strongly absorptive regime (red) located below the data points from weakly absorptive regime (blue).

[Figure]

Figure 7: The standard deviations of the ratios were added as the dashed line. The differences in activation rates of $N_{CCN}/N_a$ and $N_d/N_{CCN}$ changed slightly throughout the LWP range. The ratios of $N_{CCN}/N_a$ range from 0.39 to 0.58 for the weakly absorptive regime and from 0.32 to 0.48 for the strongly absorptive regime. The ratios of $N_d/N_{CCN}$ range from 0.58 to 0.86 for the weakly absorptive regime and from 0.47 to 0. 64 for the strongly absorptive regime.

[Figure]

Figure 8: The top panel related to re as a function of $N_a$ has been excluded in the revision following the suggestion of Reviewer #2. For the LWP bin of 0-50 $gm^{-2}$, the $ACI_r$ values are 0.26 and 0.21 for the weakly and the strongly absorptive regimes, respectively. For the LWP bin of 200-250 $gm^{-2}$, the $ACI_r$ values are 0.13 and 0.12 for the weakly and the strongly absorptive regimes, respectively. The differences in $ACI_r$ between the two regimes and the damping of $ACI_r$ with higher LWPs are still evident.

[Figure]

Figure 9: The mean rCRE for strongly absorptive regime changed from 0.72 to 0.73 while the mean value for weakly absorptive regime didn't change.

[Figure]

**General Comments:**

1. The uncertainties and hypotheses are mentioned in the article but a discussion about them is needed. For the hypotheses, it is referred that σx is assumed constant at 0.38 (page 5 line 23): Is there a way to estimate the impact from a variation of this value according to a sensible range? The same comment goes for the supersaturation fixed at 0.2% (page 6 line 21), what are the impacts on the results if SS =

1.15%. The parameters are associated with uncertainties, as mentioned several times in the text: LWP (page 5 line 9), re and Nd (page 5 line 25), SW fluxes (page 7 line 17), the contamination by insect (page 4 line 27). Unfortunately, these uncertainties are not considered in the study when comparing the ACI parameters for the different regimes. What are the impacts of the uncertainties on the results? Can you estimate the impacts on ACI to ensure that the observed difference is real?

Thanks for the comments and suggestions.

Dong et al. (1997) did a sensitivity study of retrieved Nd from different inputs, such as LWP, solar flux and $\sigma_x$. With a change of $\sigma_x\pm0.15$, Nd values vary 15.7% and 30.4%, respectively. Considering the Nd retrieval depends on multiple variables, only counting the uncertainty of one parameter $\sigma_x$ in retrieving $N_d$ cannot be representative to the actual $N_d$ uncertainty. Therefore, considering an average uncertainty of $N_d$ as 25% from the comparison with in-situ measurement (Dong et al., 1998) should be more appropriate. In that case the activation range for the weakly absorptive aerosol regime is 52% ~ 86%, while for the strongly absorptive aerosol regime is 41% ~ 67%. Therefore, based on the discussion above we decided to remove the sentence of 'Moreover, it is noteworthy that the uncertainty in deriving the CCN activation rate...'.

To test the uncertainty of CCN activation rate under different supersaturation levels, the CCN number concentration was interpolated for the 1.15% SS level using the same method described in the manuscript. As a result, with a range of SS values from 0.2% to 1.15%, the ratios of Nd/$N_{CCN}$ for weakly absorptive regime range from 0.54 to 0.38, while for strongly absorptive regime range from 0.45 to 0.25. In a given supersaturation level, the differences in CCN activation rate between two regimes do exist. In the continental boundary layer stratus, it is very hard to reach a supersaturation level of 1.15%. Therefore, we chose the SS level of 0.2% in this study to represent the most typical condition for this kind of cloud.

Based on the sensitivity study, the 10% changes of cloud LWP and downward SW at the surface would result in the 10% uncertainty in $r_e$ retrieval (Dong et al., 1997). When compared with aircraft in situ measurements, the differences between retrievals and in situ measurements are around 10% (Dong et al. 1998 and 2002). Therefore, to assess the impact of $r_e$ uncertainty on $ACI_r$, we placed the anthropogenic perturbations within the corresponding uncertainty ($\pm10\%$) range onto $r_e$ and recalculated the additional regression fits (dotted lines) for each regime in Figure 8. As a result, for that 10% change in $r_e$, the change in the logarithmic slopes ($ACI_r$) is almost negligible, which indicates that the impact of $r_e$ uncertainty on $ACI_r$ is minor and the observed differences do exist. Accordingly, the discussion above has been added to the second paragraph of section 3.3.6 in the revised manuscript.

2. Two "meteorological" parameters have been considered (section 2.3): the liquid water potential temperature and the total water mixing ratio. These parameters have been used to consider if the boundary layer is well-mixed or not. I think a deeper study on the impact of meteorological parameters on ACI would increase the impact of the paper. As shown in Table 1, the cases correspond to different seasons and airmass sources. The different ACI observed for absorptive or non-absorptive aerosols might be due to different meteorological parameters (as stated in page 11 line 16) and potentially not to the difference of aerosol optical properties. I think different regimes based on meteorology parameters (e.g., stability) should be considered to strengthen the results.

Thanks for the comments and suggestions.

To examine the influence of meteorological factors, the Lower Tropospheric Stability (LTS), which is defined as the potential temperature difference between surface and 700hPa, is used to investigate the difference in large-scale thermodynamic condition. The LTS is obtained from the ECMWF model

output which specifically provides for analysis at the ARM SGP site. The value is obtained by averaging over a grid box of 0.56*0.56° which is centered at SGP. The original temporal resolution of LTS is 1-hour and is then interpolated to 5-min to match the other variables, assuming the large-scale forcing would not have significant changes during every 1-hour window. Accordingly, the above description of LTS dataset has been added to the revised section 2.3 - 'Boundary Layer Condition and Lower Tropospheric Stability' in the revised manuscript.

[Figure]

As shown in Figure (a), the weakly absorptive regime is generally observed in a high LTS environment, given by a higher mean value and the distribution of LTS for the weakly absorptive regime is more negatively skewed than for the strongly absorptive regime. The LTS is largely impacted by the potential temperature difference throughout the mixed layer and if a strong temperature inversion that caps the boundary layer is present, it will result in high LTS values and in turn, a well-mixed boundary layer (Wood et al., 2006). Furthermore, Figure (b) shows LTS values sorted by LWP for two regimes and attempts to rule out the LWP dependence on LTS. For each LWP bin, the weakly absorptive regime has higher LTS value than the strongly absorptive regime. Such results indicate that even under similar available moisture conditions, the more sufficient turbulence can transport the below-cloud moisture as well as the CCN that activated from weakly absorbing aerosols into the cloud more efficiently, and thus enhance the sensitivity of cloud droplets to aerosol loading.

However, the LTS emphasizes a general thermodynamic condition in the lower troposphere with a wider domain as compared to the single-point measurement. The influence of cloud dynamics, presumably cloud-base updraft, is not negligible, since the sensitivity of cloud droplet to aerosol loading is enhanced with increasing updraft velocity as reported in previous studies (e.g., Feingold et al., 2003; McComiskey et al., 2009).

Furthermore, the radiative effect of light-absorbing aerosols on the cloud environment also cannot be neglected, since the strongly light-absorbing aerosols can absorb the solar radiation and heat the in-cloud atmosphere by emission. This perturbation of temperature structure results in the reduction of supersaturation in the cloud layer (Bond et al., 2013; Wang et al., 2013), and eventually dampens the sensitivity of cloud droplets to strongly light-absorbing aerosols.

In general, the results indicate that the $ACI_r$ can be counteracted by the absorbing aerosol radiative effect and be enhanced under a thermodynamic environment of high static stability, especially under lower LWP condition.

Accordingly, the discussion above has been added to the last paragraph of revised section 3.3.4 in the revised manuscript.

3. In the study, I do not understand if each day is taken separately to perform the analysis or if the study considers each measurements: For example, in Figure 4, we observe that some days have a large range of AE (2011/05/13), did the distribution of AE shown in Figure 2-b consider each point from Figure 4 or the average for each day (total of 16 points)? I assume that it is each point but the text needs to make it clear. Therefore, I do not understand why the cloud lifetime needs to be more than 3 hours (page 7 line 26) if each measurement is considered independently

Thanks for the comments.
The analysis was performed considering each 5-min temporal resolution data point so that the AE distribution in Figure 2b includes every point from Figure 4. For clarification, a sentence 'The probability density functions (PDFs) of aerosol and cloud properties from all 16 cases are shown in Fig. 2, note that the distributions include each of the 5-min data points.' was added to the first paragraph of section 3.1 in the revised manuscript.

The $r_e$ retrieval involves the solar transmission (Dong et al., 1997 and 1998) so that an overcasting cloud condition is required to avoid the impact of broken clouds with leakage of direct solar radiation on the transmission calculation, which is reflected in the point-based cloud radar observation as a long-lasting continuous cloud layer. Therefore, the criterion of 3-hour is a good balance between the number of cloud cases and the feasibility and stability of the retrieval.

Specific Comment:
There is no indication of how many data points are considered in the analysis.

Thanks for the comments. A total of 693 data points has been used in this study, and the detail of the number of data points used in every case was added in the revised Table 1. For clarification, the information of the number of data points has been added to the sentence 'Note that all the variables used in the study are averaged in 5-min temporal resolution bins. A total of 16 cases were selected during the 6-year period from 2007 to 2012, which represents a total of 693 samples (~ 58 hours) in this study, the detailed time period and the number of sample points of each case are listed in Table 1.' in section 2.5 in the revised manuscript.

In addition, to give the information of the number of data points that are categorized in two regimes, a sentence 'Within the 693 selected samples, 360 data points are classified in the weakly absorptive aerosol regime, while the remaining data points are in the strongly absorptive aerosol regime' has been added to the second paragraph of section 3.3.1 in the revised manuscript.

Furthermore, in the revised Figure 5d, the number of data points in every LWP bin is denoted by the numbers above every PDF bar for the two absorptive regimes. For clarification, the sentence 'The numbers above the bars in LWP distribution (Fig. 5d) for the two absorptive regimes denote the number of data points which will be used in the analysis with binned LWP in the later sections' has been added in the first paragraph of section 3.3.2. in the revised manuscript.

Abstract: A sentence about the context, and why it is important to study the aerosol cloud interaction is missing.

Thanks for the comments, a sentence of 'Aerosol indirect effect on cloud microphysical and radiative properties is one of the largest uncertainties in climate simulations. In order to investigate the aerosol-cloud interactions, a total of 16 low-level stratus cloud cases under daytime coupled boundary layer conditions are selected.', and a sentence of 'The impact of the aerosols with different light-absorbing

abilities on the sensitivity of cloud microphysical responses is also investigated' have been added to the revised abstract.

page 3 line 10: I suggest to remove the "co-albedo" has it can confuse a reader which is not familiar with this term, or to specify that it is 1-SSA.

Thanks for the suggestion.
The sentence has been changed to 'Alternatively, the single scattering albedo (SSA) and co-albedo (1-SSA) can be used to better separate the aerosol types because they focus on the relative absorbing ability of aerosols at specific wavelengths' in the third paragraph of introduction in the revised manuscript.

Page 4 line 20: The study uses two different instruments with different spatial and temporal resolutions, how does it affect your results? Is the uncertainty from the KAZR lower?

Thanks for the comment.
The uncertainty of KAZR (~30m) is lower than MMCR (~45m). The difference of 15m between these two cloud radars would not cause a significant difference in detecting the cloud boundaries, thus it would not affect the results.

page 5 line 4: Why has the "cloud-top height lower than 3 km" limit been chosen?

This limit is chosen following the definition of single-layered low cloud in Dong et al. (2006), which characterized by clouds that have cloud top height less than 3km with no clouds above them. This definition is also consistent to the ISCCP defined low clouds (> 680 mb).

page 6 line 14: What is the initial temporal resolution?

The initial temporal resolution is 1 minute and then the data were averaged into 5 minutes to match other variables. For clarification, the sentence has been changed to 'In this study, the sub-10 μm aerosol optical properties with original 1-min temporal resolution were averaged into 5-min bins to match the cloud microphysical properties' in the first paragraph of section 2.2 in the revised manuscript.

Is there a study comparing the measurements from SGP with in-situ data to evaluate the performance of the instruments? The results are provided for cloud microphysical properties (page 5 line 25), but is there something similar for the aerosol properties, cloud boundaries, and boundary layer conditions?

Yes, a study was conducted by Delle Monache et al., (2004) used in-situ aerosol measurements from 59 flights during March 2000 – March 2001 to compare with the surface aerosol measurements. Their results showed that the aerosol extensive properties measured within the boundary layer were well-correlated with surface measurements. Thus, under the well-mixed cloud-topped boundary layer condition, the surface measurements of aerosol properties are well representative of the boundary layer aerosols which actually influence the cloud microphysical properties.

page 7 line 6: 0.5 K and 0.5 g/kg: are these thresholds the same as in Dong et al. (2015)?

Yes, these thresholds are the same as in Dong et al., 2015, originally suggested by Jones et al. (2011).

page 8 line 24: Can you describe the difference between FMF and AE? I am not sure to understand why the study needs the two parameters.

The AE focuses on the relative difference of scattering abilities in a specific aerosol group (that belong to the size category of < 1μm or < 10μm) at two different wavelengths, which reveal the relative wavelength dependence of particle optical properties due to differences in particle sizes. But it can intrinsically carry uncertainty if the mixtures of different size aerosols share similar spectral dependences. While the FMF focuses on one single wavelength and describes the aerosol scattering ability at this wavelength, given by the ratio of the fine-mode (diameter < 1μm) aerosol scattering coefficient to the total (diameter <10μm) aerosol scattering coefficient ($\sigma_{sp1}/\sigma_{sp10}$), which pertains to the relative contribution of fine-mode aerosol scattering in total scattering. The use of the FMF parameter along with AE can give a robust illustration of the fine-mode aerosol dominance in the selected cloud cases.

page 10 line 5: How is the uncertainty on ACI retrieved? Is it the 95% confidence interval of the fit?

Yes, the uncertainty of ACI is retrieved from the 95% confidence interval of the fit

page 10 line 6-: The authors are comparing ACI values with previous studies. I am a bit skeptical about it: There is plenty of studies retrieving ACI values with different methods, datasets, geographical locations. ACI parameter depends on that. The authors only report ACI values which range with their study without a discussion on the differences. I think there is two different possibilities: Either you consider all the studies retrieving ACI and discuss about the potential differences or the comparison is limited to studies looking at the same region and/or same data.

Thanks for the comment and suggestion. The discussion about previous studies has been confined to the studies were carried out with respect to the low-level stratiform clouds over the SGP site only. The differences in the sampling of aerosol and cloud properties, as well as the conditional dependences of $ACI_r$, examined in every study were included in the revised discussion, to better understand the influence of different factors on the assessment of $ACI_r$.

Accordingly, this discussion in the last paragraph of section 3.2 in the revised manuscript has been changed to:
'At the ARM-SGP site, based on the analysis on seven selected stratocumulus cases during the period 1998 - 2000, Feingold et al. (2003) reported the first ground-based measured $ACI_r$ values of 0.02 to 0.16 using the lidar measured aerosol extinction at a wavelength of 355 nm as the proxy for aerosol loading. The data were stratified in similar LWP bins to eliminate the LWP effect on $r_e$. The study conducted by Feingold et al. (2006) during an intensive operation period in May 2003 showed that the assessment of $ACI_r$ can be affected by the usage of different aerosol proxies and boundary layer conditions. Using surface measured $N_a$ to represent aerosol loading yielded unrealistic values of $ACI_r$ even after sorted by LWP, presumably owing to decoupled boundary layer conditions. However, if the surface aerosol scattering coefficient ($\sigma_{sp}$) and aerosol extinction at an altitude of 350 m are used as CCN proxies, then similar $ACI_r$ values can be obtained with a range of 0.14-0.39. Under coupled conditions, the $N_a$ and $\sigma_{sp}$ could serve as reliable CCN proxies. The $\sigma_{sp}$ of accumulation-mode aerosols was used in Kim et al. (2008) to show that the $ACI_r$ can be better manifested in the adiabatic cloud than in sub-adiabatic environment, despite the relatively lower values (0.04 – 0.17) retrieved in stratus cloud cases during the period 1999 -2001. Moreover, this influence of thermodynamic condition on $ACI_r$ was further documented in Kim et al. (2012) where the aerosol-cloud interaction found to be enhanced under the condition of strong inversion above the stratus layer.'

page 11 lines 23-26: I do not understand the sentence, can you rephrase it?

Thanks for the suggestion, the sentence has been changed to 'The distributions of $N_a$ from the two absorptive regimes is comparable to one another. The mean $N_{CCN}$ for the weakly absorptive regime (559 cm$^{-3}$) is larger than that from the strongly absorptive regime (384 cm$^{-3}$), and the occurrence of high $N_{CCN}$ values (larger than 1000 cm$^{-3}$) is also higher in the weakly absorptive regime' in the first paragraph of section 3.3.2 in the revised manuscript.

page 11 line 26-27: Can it be quantified?

Thanks for the comment.
Unfortunately, since the value of AE is retrieved via a logarithmic slope, the AE is emphasizing the relative dominance of fine-mode or coarse-mode aerosols within an aerosol plume rather than the absolute amount of existence. Generally, the AE > 1 indicates the particle size distributions dominated by fine mode aerosols (submicron), and AE < 1 denotes the dominance of coarse mode aerosols. Thus, the dominance of fine-mode aerosol is hard be quantified based on the value of AE.

page 12 line16: The "majority", can it be quantified?

Thanks for the comment. The sentence has been changed to 'For a broad range of $N_a$, especially 200-700 cm$^{-3}$ and 1200-3500 cm$^{-3}$, the majority (~74%) of sample points from the strongly absorbing regime are located below the samples from the weakly absorbing regime' in the first paragraph of section 3.3.3 in the revised manuscript.

Figure 2: Can the standard deviation be displayed with the mean? Also, considering that the distributions are not Gaussian, why did you consider the mean rather the median?

Thanks for the comments. We totally agree that the median value can better represent a non-normal distribution. The mean values were originally used considering some variables in Figure 2 were normally distributed. Therefore, in the revised Figure 2, the standard deviations are now displayed with the mean, and the median values of the variables are also displayed to better represent the data distributions.

Figure 7: Is there a reason why the standard deviation is not included for the ratios NCCN to Na and Nd to NCCN?

The standard deviations of the two ratios were originally not included because of the consideration of a better viewing of the figure.
The standard deviations of the ratios are now displayed as the dashed line in revised Figure 7.

Technical corrections:

page 5 line 18 "100" should not be here.

Thanks for the comment, the "100" here denote the value of LWP in a unit of gm$^{-3}$ should be multiplied by 100 in the re retrieval algorithm (Dong et al., 1998).

page 8 line 17: find → fine

Thanks for pointing out, the correction has been made in the revised manuscript.

page 15 line 25: Fig. 10 → Fig. 8

Thanks for pointing out, the correction has been made in the revised manuscript.

Figure2 caption: the order to describe the figures is: a-d-b-c-. . . instead of a-b-c-d-. . .

Thanks for pointing out, the order has been changed alphabetically in the revised Figure 2 caption.

Figure 4: The definitions of the dotted lines are missing in the caption?

Thanks for pointing out, a description 'Horizontal dotted line denotes the demarcation of $AE_{450-700nm} = 1$; Vertical dotted line denote the demarcation of $\omega_{abs450} = 0.07$.' has been added to the revised Figure 4 caption.

**References**

Bond, T. C., Doherty, S. J., Fahey, D. W., Forster, P. M., Berntsen, T., Deangelo, B. J., Flanner, M. G., Ghan, S., Kärcher, B., Koch, D., Kinne, S., Kondo, Y., Quinn, P. K., Sarofim, M. C., Schultz, M. G., Schulz, M., Venkataraman, C., Zhang, H., Zhang, S., Bellouin, N., Guttikunda, S. K., Hopke, P. K., Jacobson, M. Z., Kaiser, J. W., Klimont,

Delle Monache, L., Perry, K. D., Cederwall, R. T., and Ogren, J. A.: In situ aerosol profiles over the Southern Great Plains cloud and radiation test bed site: 2. Effects of mixing height on aerosol properties, J. Geophys. Res., 109, D06209, doi:10.1029/2003JD004024, 2004.

Dong, X., Ackerman, T. P., Clothiaux, E. E., Pilewskie, P. and Han, Y.: Microphysical and radiative properties of boundary layer stratiform clouds deduced from ground-based measurements, J. Geophys. Res. Atmos., 1997.

Dong, X., Ackerman, T. P. and Clothiaux, E. E.: Parameterizations of the microphysical and shortwave radiative properties of boundary layer stratus from ground-based measurements, J. Geophys. Res. Atmos., doi:10.1029/1998JD200047, 1998.

Dong, X., Minnis, P., Mace, G. G., Smith, W. L., Poellot, M., Marchand, R. T. and Rapp, A. D.: Comparison of stratus cloud properties deduced from surface, GOES, and aircraft data during the March 2000 ARM cloud IOP, J. Atmos. Sci., doi:10.1175/1520-0469(2002)059<3265:COSCPD>2.0.CO;2, 2002.

Dong, X., Xi, B. and Minnis, P.: A climatology of midlatitude continental clouds from the ARM SGP Central Facility. Part II: Cloud fraction and surface radiative forcing, J. Clim., doi:10.1175/JCLI3710.1, 2006.

Dong, X., Schwantes, A. C., Xi, B. and Wu, P.: Investigation of the marine boundary layer cloud and CCN properties under coupled and decoupled conditions over the azores, J. Geophys. Res., doi:10.1002/2014JD022939, 2015.

Feingold, G., Eberhard, W. L., Veron, D. E. and Previdi, M.: First measurements of the Twomey indirect effect using ground-based remote sensors, Geophys. Res. Lett., doi:10.1029/2002GL016633, 2003.

Feingold, G., Furrer, R., Pilewskie, P., Remer, L. A., Min, Q. and Jonsson, H.: Aerosol indirect effect studies at Southern Great Plains during the May 2003 Intensive Operations Period, J. Geophys. Res. Atmos., doi:10.1029/2004JD005648, 2006.

Jones, C. R., Bretherton, C. S. and Leon, D.: Coupled vs. decoupled boundary layers in VOCALS-REx, Atmos. Chem. Phys., doi:10.5194/acp-11-7143-2011, 2011.

Kim, B. G., Miller, M. A., Schwartz, S. E., Liu, Y., and Min, Q.: The role of adiabaticity in the aerosol first indirect effect, J. Geophys. Res., 113, D05210, doi:10.1029/2007JD008961, 2008.

Kim, Y. J., Kim, B. G., Miller, M., Min, Q. and Song, C. K.: Enhanced aerosol-cloud relationships in more stable and adiabatic clouds, Asia-Pacific J. Atmos. Sci., doi:10.1007/s13143-012-0028-0, 2012.

McComiskey, A, Feingold, G., Frisch, A. S., Turner, D. D., Miller, M., Chiu, J. C., Min, Q., and Ogren, J.: An assessment of aerosol-cloud interactions in marine stratus clouds based on surface remote sensing, J. Geophys. Res., 114, D09203, doi:10.1029/2008JD011006, 2009.

Wang, Y., Khalizov, A., Levy, M. and Zhang, R.: New Directions: Light absorbing aerosols and their atmospheric impacts, Atmos. Environ., doi:10.1016/j.atmosenv.2013.09.034, 2013.

Wood, R. and Bretherton, C. S.: On the relationship between stratiform low cloud cover and lower-tropospheric stability, J. Clim., doi:10.1175/JCLI3988.1, 2006.

---

## Author Comment (AC2) · 22 Sep 2019

**Response to Reviewer #2**

We appreciate your time for carefully reviewing our manuscript. We would like to thank you for the constructive comments and suggestions, which encourage and help us to improve the manuscript. The manuscript has been revised accordingly. In the response below, the reviewer's comments are provided in black text and our responses are provided in blue text.

**Response:**

General: Temporal resolution and number of data points While the temporal resolution of some of data products used to create the data set is given (I think everything is averaged to 5-minutes but it's not explicitly noted anywhere), nowhere is it stated the number of data points used to determine the ACIr index and other correlations. The 16 cloud cases and total time that the data set covers is provided in Table 1 but does not contain these statistics, which are important for interpreting results. If I do some math it seems that there are sufficient statistics but text needs to be added to fully and clearly describe the statistics of the data set for the reader.

Thanks for the comments. A total of 693 data points has been used in this study, and the detail of the number of data points used in every case was added in the revised Table 1. For clarification, the information of the number of data points has been added to the sentence 'Note that all the variables used in the study are averaged in 5-min temporal resolution bins. A total of 16 cases were selected during the 6-year period from 2007 to 2012, which represents a total of 693 samples (~ 58 hours) in this study, the detailed time period and the number of sample points of each case are listed in Table 1' in section 2.5 in the revised manuscript.

In addition, to give the information of the number of data points that are categorized in two regimes, a sentence 'Within the 693 selected samples, 360 data points are classified in the weakly absorptive aerosol regime, while the remaining data points are in the strongly absorptive aerosol regime' has been added to the second paragraph of section 3.3.1 in the revised manuscript.

Furthermore, in the revised Figure 5d, the number of data points in every LWP bin is denoted by the numbers above every PDF bar for the two absorptive regimes. For clarification, the sentence 'The numbers above the bars in LWP distribution (Fig. 5d) for the two absorptive regimes denote the number of data points which will be used in the analysis with binned LWP in the later sections' has been added in the first paragraph of section 3.3.2. in the revised manuscript.

Choice of aerosol data used - The authors choose to use the sub-micron aerosol optical properties from the available measurements rather than the sub-10 um. What is the motivation for this choice? The total aerosol number concentration Na and CCN are used which are not restricted to the sub-micron size cut. It's not fully consistent that the Na and CCN be sorted by high and low absorbing regime according to the sub-micron absorption – there could be a relationship between size and composition/absorption. Further, the sub-micron scattering fraction is presented alongside the scattering angstrom exponent for the sub-micron aerosol

only. This lacks consistency and can make interpretation of the results difficult when reading through progressive steps in the analysis. An explanation (and implications) for how the sub-micron only properties relate to the others should be given if the choice of data is not changed. It probably won't change the overall picture given that the aerosol is largely sub-micron but why complicate the issue?

Thanks for the comments and suggestions. The original choice of submicron aerosol data was due to the consideration of fine-mode aerosol dominance over SGP. However, we totally agree that caused an inconsistency in aerosol number concentration and optical properties. Therefore, aerosol optical properties are based on measurements of the sub-10μm size-cut in the revised manuscript and the data were re-sorted by the (1-SSA) values of sub-10μm aerosols into the weakly and strongly absorptive regimes, accordingly. In general, the main results and conclusions did not change significantly given the fact that fine-mode aerosols dominated the aerosol plumes over SGP.

The Figures below show the corresponding changes after new aerosol results have been used in the revised version:

Figure 2: For the total dataset, the mean value of AE changes from 1.67 to 1.57; the mean value of SSA changes from 0.94 to 0.93, which results from the contribution of aerosols having a range of diameters from 1 – 10 microns.

[Figure]

Figure 4: As indicated in the revised Figure 2, the values of (1-SSA) generally increase and AE values generally decrease owing to the inclusion of aerosols having diameters greater than 1 micron.

[Figure]

Figure 5: The mean values changed slightly due to the new categories of absorptive regimes, but the differences in distributions and mean values between the two regimes were preserved.

[Figure]

Figure 6: The overall activation rates did not change, with more data points from the strongly absorptive regime (red) located below the data points from weakly absorptive regime (blue).

[Figure]

Figure 7: The standard deviations of the ratios were added as the dashed line. The differences in activation rates of $N_{CCN}/N_a$ and $N_d/N_{CCN}$ changed slightly throughout the LWP range. The ratios of $N_{CCN}/N_a$ range from 0.39 to 0.58 for the weakly absorptive regime and from 0.32 to 0.48 for the strongly absorptive regime. The ratios of $N_d/N_{CCN}$ range from 0.58 to 0.86 for the weakly absorptive regime and from 0.47 to 0. 64 for the strongly absorptive regime.

[Figure]

Figure 8: The top panel related to re as a function of $N_a$ has been excluded in the revision following the suggestion of Reviewer #2. For the LWP bin of 0-50 $gm^{-2}$, the $ACI_r$ values are 0.26 and 0.21 for the weakly and the strongly absorptive regimes, respectively. For the LWP bin of 200-250 $gm^{-2}$, the $ACI_r$ values are 0.13 and 0.12 for the weakly and the strongly absorptive regimes, respectively. The differences in $ACI_r$ between the two regimes and the damping of $ACI_r$ with higher LWPs are still evident.

[Figure]

Figure 9: The mean rCRE for strongly absorptive regime changed from 0.72 to 0.73 while the mean value for weakly absorptive regime didn't change.

[Figure]

Lack of meteorological parameters or aerosol radiative effects in assessing covariances of aerosol and cloud properties - My greatest concern with this analysis relates to the association of aerosol absorption to cloud microphysics and cloud radiative effect without considering meteorological or systematic seasonal influences that may be affecting the co-variance of aerosol and cloud properties. The relationship of Na to CCN for high and low absorption regimes is compelling and it does seem that the difference in composition has an effect on the number of CCN. But then the examination of the relationship between CCN and drop number is presented without any discussion of controls by cloud dynamics or potential radiative effects of the absorbing aerosol on the environment of cloud dynamics.

Thanks for the comments and suggestions. We totally agree that both the meteorological factors and aerosol radiative effect could have a non-negligible influence on the aerosol-cloud interaction.

To examine the influence of meteorological factors, the Lower Tropospheric Stability (LTS), which is defined as the potential temperature difference between surface and 700hPa, is used to investigate the difference in large-scale thermodynamic condition. The LTS is obtained from the ECMWF model output which specifically provides for analysis at the ARM SGP site. The value is obtained by averaging over a grid box of 0.56*0.56° which is centered at SGP. The original temporal resolution of LTS is 1-hour and is then interpolated to 5-min to match the other variables, assuming the large-scale forcing would not have significant changes during every 1-hour window. Accordingly, the above description of LTS dataset has been added to the revised section 2.3 - 'Boundary Layer Condition and Lower Tropospheric Stability' in the revised manuscript.

[Figure]

As shown in Figure (a), the weakly absorptive regime is generally observed in a high LTS environment, given by a higher mean value and the distribution of LTS for the weakly absorptive regime is more negatively skewed than for the strongly absorptive regime. The LTS is largely impacted by the potential temperature difference throughout the mixed layer and if a strong temperature inversion that caps the boundary layer is present, it will result in high LTS values and in turn, a well-mixed boundary layer (Wood et al., 2006). Furthermore, Figure (b) shows LTS values sorted by LWP for two regimes and attempts to rule out the LWP dependence on LTS. For each LWP bin, the weakly absorptive regime has a higher LTS value than the strongly absorptive regime. Such results indicate that even under similar available

moisture conditions, the more sufficient turbulence can transport the below-cloud moisture as well as the CCN that activated from weakly absorbing aerosols into the cloud more efficiently, contributing to a higher conversion rate of $N_d/N_{CCN}$ in the weakly absorptive regime.

However, the LTS emphasizes a general thermodynamic condition in the lower troposphere with a wider domain as compared to the single-point measurement. The influence of cloud dynamics, presumably cloud-base updraft, is not negligible, since the sensitivity of cloud droplet to aerosol loading is enhanced with increasing updraft velocity as reported in previous studies (e.g., Feingold et al., 2003; McComiskey et al., 2009).

Furthermore, the radiative effect of light-absorbing aerosols on the cloud environment also cannot be neglected, since the strongly light-absorbing aerosols can absorb solar radiation and heat the in-cloud atmosphere by emission. This perturbation of temperature structure results in the reduction of supersaturation in the cloud layer (Bond et al., 2013; Wang et al., 2013), and eventually dampens the conversion process from CCN to cloud droplet.

Unfortunately, due to the lack of measurement of cloud-base vertical velocity throughout the studying period, this competing effect of cloud thermodynamic and dynamic cannot be fully untangled from the aerosol effect given the currently available dataset. The differences in conversion rates of Nd/Nccn between the two regimes might be affected by the combined effects of LTS, updraft velocity, and aerosol absorption effect on the cloud environment.

Accordingly, the discussion above has been added to the last paragraph of revised section 3.3.4 in the revised manuscript.

Related is the fact that most cases occurred only during winter and spring and largely under northerly wind conditions. Also, the authors state that the high and low absorption cases split largely along the same lines with higher absorption occurring in spring, however the implications of the co-variability in aerosol and cloud properties is never discussed. You note that the LWP is larger under the high aerosol absorption regime – is this causal? A seasonal effect? You also note that higher absorption occurs in Spring. This fact is not revisited and explained in the discussion after all relationships have been analyzed. These two factors could be unrelated but both driven by seasonal effects on aerosol distributions and available moisture separately. What implications would this have for the relationships you present here?

[Figure]

Thanks for the comments and suggestions.

The figure above shows the seasonal variation of LWP for single-layered low clouds during the period 2007-2012. Note that the mean value of LWP in Spring (149 $gm^{-2}$) is slightly higher than that in Winter (138 $gm^{-2}$). Similar results (LWP=160 vs. 141.1 $gm^{-2}$) are found in Dong et al. (2005) who used the same dataset but for different period (1997-2002). In Spring, owing to the upper-level ridge centered over the western Atlantic, the SGP is located at the northwest edge of the Sub-tropical High. Therefore, the SGP during the spring months is under the influence of relatively frequent southerly transport, which is characterized by strongly absorbing carbonaceous aerosols produced from biomass burning from Central America, as well as the moisture transported from the Gulf of Mexico. While during Winter, the SGP site experiences airmasses from higher latitudes with less intrusion of airmasses from the south (Andrews et al., 2011; Parworth et al., 2015; Logan et al., 2018).

The seasonal differences in aerosol distributions and available moisture between the two absorptive regimes are largely due to the different airmass transport pathways induced by the seasonal synoptic patterns, and no clear causality is found between springtime higher LWP and absorbing aerosols. In addition, the analyses of aerosol-cloud interaction in the manuscript are performed by stratified LWP, which eliminates the effect of different LWPs on the aerosol and cloud properties.

Accordingly, the following discussion has been added to the last paragraph of section 3.3.1 in the revised manuscript:

'In spring, owing to the upper-level ridge centered over the western Atlantic, the SGP is located at the northwestern edge of the sub-tropical high. Under this synoptic pattern, the SGP is under the influence of relatively frequent southerly transport of the airmasses from Central America, which is characterized by strongly absorbing carbonaceous aerosols produced from biomass burning, as well as the moisture transported from the Gulf of Mexico. During the winter, the SGP site experiences the transported airmasses from higher latitudes with less intrusion of airmasses from the south (Andrews et al., 2011; Parworth et al., 2015; Logan et al., 2018)'.

And the following statement has been added to section 3.3.2 in the revised manuscript:

'This LWP difference might be associated with the seasonality of airmass transport over the SGP as discussed in section 3.3.1. Although the seasonality of aerosol distribution and LWP have similar trends, no clear causality has been found between them.'

In Fig 8, why look at drop effective radius as a function of Na if you have CCN measurements and have already established the Na to CCN relationship dependence on absorption? I feel like the effect of size and composition on drop activation are getting conflated here and in some other places in the manuscript. Given that the absorption dependence of Na to CCN is compelling, you might to better to simplify the paper by omitting some of these plots that don't add to the message and can actually be confusing. On P16, the last paragraph of section 3.3.6 has a related discussion that is confusing. The relationship of Na to clouds and CCN to clouds is considered. This doesn't make sense – the definition of CCN is the segment of the total aerosol population that will activate to form cloud drops (the statement 'clouds are more sensitive to CCN than solely aerosol particles' should be deleted.) In reality the number of cloud drops might not equal CCN, but if the measurements are good then that is due to some competing effect of cloud dynamics, available moisture, radiative effects, etc. (none of the latter are addressed here.) Other statements in the paper that follow this confusing logic are P18 L4-5 '. . .conversion rates of Nd/Nccn for weakly absorbing aerosols are higher than for strongly absorbing aerosols' suggests that there is some other mechanism at play like a radiative effect – or the CCN measurement is not accurate. Also P18 L13-14 '..the mechanism from CCN to cloud droplet is more straightforward than from aerosol particle to cloud droplet.' It ought to be.

Thanks for the comments and suggestions.
We totally agree that the current discussion about the relationship of CCN to cloud droplets conveys confusing messages. Therefore, we have deleted the related discussions in the revised manuscript.

More specifically, the last paragraph of section 3.3.6 in the revised manuscript has been modified to:
'Note that the LTS values from the weakly absorptive regime (22.91K and 19.78K) are higher than those from the strongly absorptive regime (21.72K and 17.83K) for the selected two LWP bins. As discussed in the previous section, on the one hand, owing to the stronger temperature inversion indicated by the higher LTS values, low clouds are more closely connected to weakly absorbing aerosols and moisture below cloud by efficient turbulence. On the other hand, with the presence of strongly light-absorbing aerosols, the cloud layer heating induced by the aerosol absorptive effect can result in the reduction of in-cloud supersaturation and leads to the damping of cloud microphysical sensitivity to strongly absorbing aerosols. In general, the results indicate that the $ACI_r$ can be counteracted by the absorbing aerosol radiative effect and be enhanced under a thermodynamic environment of high static stability, especially under lower LWP conditions.'

Misc: The discussion at the top of P10 regarding results from other studies that calculated ACI should include the parameters and sampling used in those studies to provide some background

on why values might differ. Some discussion of the results rather than a simple reporting of the numbers should be included. How the data is sorted and what dependencies are examined can have a large impact on this indexes due to the inherent sensitivity of cloud microphysics to a range of parameters.

Thanks for the comment and suggestion. The discussion about previous studies has been confined to the studies were carried out with respect to the low-level stratiform clouds over the SGP site only. The differences in the sampling of aerosol and cloud properties, as well as the conditional dependences of $ACI_r$, examined in every study were included in the revised discussion, to better understand the influence of different factors on the assessment of $ACI_r$.

Accordingly, this discussion in the last paragraph of section 3.2 in the revised manuscript has been changed to:
'At the ARM-SGP site, based on the analysis on seven selected stratocumulus cases during the period 1998 - 2000, Feingold et al. (2003) reported the first ground-based measured $ACI_r$ values of 0.02 to 0.16 using the lidar measured aerosol extinction at a wavelength of 355 nm as the proxy for aerosol loading. The data were stratified in similar LWP bins to eliminate the LWP effect on $r_e$. The study conducted by Feingold et al. (2006) during an intensive operation period in May 2003 showed that the assessment of $ACI_r$ can be affected by the usage of different aerosol proxies and boundary layer conditions. Using surface measured $N_a$ to represent aerosol loading yielded unrealistic values of $ACI_r$ even after sorted by LWP, presumably owing to decoupled boundary layer conditions. However, if the surface aerosol scattering coefficient ($\sigma_{sp}$) and aerosol extinction at an altitude of 350 m are used as CCN proxies, then similar $ACI_r$ values can be obtained with a range of 0.14-0.39. Under coupled conditions, the $N_a$ and $\sigma_{sp}$ could serve as reliable CCN proxies. The $\sigma_{sp}$ of accumulation-mode aerosols was used in Kim et al. (2008) to show that the $ACI_r$ can be better manifested in the adiabatic cloud than in sub-adiabatic environment, despite the relatively lower values (0.04 – 0.17) retrieved in stratus cloud cases during the period 1999 -2001. Moreover, this influence of thermodynamic condition on $ACI_r$ was further documented in Kim et al. (2012) where the aerosol-cloud interaction found to be enhanced under the condition of strong inversion above the stratus layer.'

In Fig 7 are the differences in the ratios statistically significant? Where standard deviations are included it's easier to judge what might be significant, but there is a general lack of discussion of uncertainties and statistical significant throughout. This is further complicated by the lack of information on the number of data points used in each analysis (or each bin in the binned analyses) as commented above.

Thanks for the comments, a student's t-test was performed to test the ratio difference in every LWP bin at the 95% significance level. The standard deviations of the ratios were plotted as the dashed line in the revised Fig 7. For clarification, the sentence 'A student's t-test is performed to test the ratio difference in each LWP bin at the 95% significance level. The results indicate the ratio differences between two absorptive regimes are statistically significant' has been added to the first paragraph of section 3.3.4 in the revised manuscript.

Figures: All of the labels need to be much bigger – many are very difficult to read. Figure 2 caption has the sub-figured listed out of order – should be ordered alphabetically from a-f Figure 3 red and orange colors are indistinguishable Figure 6 not points above the 1:1 line is curious – this almost always exists due to measurement error – were these removed?

Thanks for the comments. The labels of the revised figures have been enlarged for better viewing. The caption of Figure 2 has been corrected following the alphabetical order. The orange color (corresponding to date 20120204) in Figure 4 has been changed to a cyan color. And yes, we considered that the sample points with higher $N_a$ value than $N_{CCN}$ value were a result of instrument error of CPC or CCN counter, thus we removed those points for better data quality.

Specific: Page and Line P2 L23: 'influence' rather that 'interact with' (suggestion)

Thanks for the suggestion, the sentence has been changed to 'The physical mechanism underlying the aerosol effect on clouds is that aerosols activate as cloud condensation nuclei (CCN) and then influence the cloud microphysical features' in the revised manuscript.

P2 L28: 'inferred' rather than 'identified by' (suggestion) – your explanation of the uses and limitations of inferring composition from optical properties is quite nice

Thanks for the suggestion, the sentence has been changed to 'Previous studies have suggested that the composition of aerosols can be inferred by their optical properties such as aerosol optical depth, single scattering albedo, and Ångström exponent' in the revised manuscript.

P3 L6-9: may also not that measurements of absorption angstrom exponent typically carry large uncertainties

Thanks for the suggestion, the sentence has been changed to 'Although studies have been done to classify aerosol types using the absorption Ångström exponent, which is associated with the absorptive spectral dependence of particles, the measurement of this parameter typically carry large uncertainty, and has limited value when there are mixtures of different aerosol species that share similar spectral dependences' in the revised manuscript.

P7 L25-27: and the restriction of LWP > 20 g m-3

Thanks for the suggestion, the sentence has been changed to '…the selection of cloud cases is limited by the following criteria: non-precipitating and cloud-top height less than 3 km with lifetime more than 3 hours under the limitation of 20 $gm^{-2}$ < LWP < 300 $gm^{-2}$ and the coupled boundary layer conditions' in the revised manuscript.

P7 L27: what is the reasoning behind the daytime only? Simply that the quantity is only available under sunlight conditions? Consider rewording

Thanks for the comments, the sentence has been changed to 'Only daytime cloudy periods were considered in this study because the $r_e$ retrieval required the information of solar transmission (Dong et al., 1998)' in the revised manuscript.

P8 L17: 'find' should be 'fine'

Thanks for pointing out, the correction has been made in the revised manuscript.

P8 L22-24: this sentence is confusing – maybe '. . .greater than 0.6 represent the dominance of fine mode aerosol in the total population and values less than 0.2 represent the dominance of coarse mode aerosols in the total population.'

Thanks for the suggestion, the sentence has been changed accordingly in the revised manuscript.

P8 L25: 'dominated' should be 'dominant'

Thanks for pointing out, the correction has been made in the revised manuscript.

P9 L16: note that 'theoretical' values of ACIr. . .

Thanks for the suggestion, the sentence has been changed to 'Note that values of ACIr have theoretical boundaries of 0-0.33…'in the revised manuscript.

P11 L9: don't think you can state that co-albedo provides information about composition, just more sensitive to the amount of absorption

Thanks for the comment, the sentence has been changed to 'This parameter is more sensitive to the capabilities of aerosol light absorption (rather than scattering) in total aerosol light extinction and therefore can better infer the aerosol composition' in the revised manuscript.

P11 L22-25: sentence needs rewriting – may just need a 'For' at the start and to remove 'higher' at the end

Thanks for the suggestion, the sentence has been changed to 'The distributions of $N_a$ from the two absorptive regimes is comparable to one another. The mean $N_{CCN}$ for the weakly absorptive regime (559 cm$^{-3}$) is larger than that from the strongly absorptive regime (384 cm$^{-3}$), and the occurrence of high $N_{CCN}$ values (larger than 1000 cm$^{-3}$) is also higher in the weakly absorptive regime' in the revised manuscript.

P15 L19-22: how much does the composition of CCN matter for growth once it's already activated?

Thanks for the comments. We found that this statement cannot be fully supported by the current analysis. Therefore, the last part of section 3.3.5 has been modified to 'The combination of cloud thermodynamic, dynamic, and aerosol radiative effects impact the conversion process from CCN to cloud droplet. Under a given moisture availability, a greater number of CCN in the weakly absorptive regime can be converted to cloud droplets. This results in higher number concentrations of smaller cloud droplets, while the lower CCN activating rate in the strongly absorptive regime leads to fewer and larger cloud droplets at a fixed LWP' in the revised manuscript.

P15 L25: should be Fig 8

Thanks for pointing out, the correction has been made in the revised manuscript.

**References**

Andrews, E., Sheridan, P. J. and Ogren, J. A.: Seasonal differences in the vertical profiles of aerosol optical properties over rural Oklahoma, Atmos. Chem. Phys., doi:10.5194/acp-11-10661-2011, 2011.

Bond, T. C., Doherty, S. J., Fahey, D. W., Forster, P. M., Berntsen, T., Deangelo, B. J., Flanner, M. G., Ghan, S., Kärcher, B., Koch, D., Kinne, S., Kondo, Y., Quinn, P. K., Sarofim, M. C., Schultz, M. G., Schulz, M., Venkataraman, C., Zhang, H., Zhang, S., Bellouin, N., Guttikunda, S. K., Hopke, P. K., Jacobson, M. Z., Kaiser, J. W., Klimont, Z., Lohmann, U., Schwarz, J. P., Shindell, D., Storelvmo, T., Warren, S. G. and Zender, C. S.: Bounding the role of black carbon in the climate system: A scientific assessment, J. Geophys. Res. Atmos., doi:10.1002/jgrd.50171, 2013.

Dong, X., Ackerman, T. P. and Clothiaux, E. E.: Parameterizations of the microphysical and shortwave radiative properties of boundary layer stratus from ground-based measurements, J. Geophys. Res. Atmos., doi:10.1029/1998JD200047, 1998.

Feingold, G., Eberhard, W. L., Veron, D. E. and Previdi, M.: First measurements of the Twomey indirect effect using ground-based remote sensors, Geophys. Res. Lett., doi:10.1029/2002GL016633, 2003.

Feingold, G., Furrer, R., Pilewskie, P., Remer, L. A., Min, Q. and Jonsson, H.: Aerosol indirect effect studies at Southern Great Plains during the May 2003 Intensive Operations Period, J. Geophys. Res. Atmos., doi:10.1029/2004JD005648, 2006.

Kim, B. G., Miller, M. A., Schwartz, S. E., Liu, Y., and Min, Q.: The role of adiabaticity in the aerosol first indirect effect, J. Geophys. Res., 113, D05210, doi:10.1029/2007JD008961, 2008.

Kim, Y. J., Kim, B. G., Miller, M., Min, Q. and Song, C. K.: Enhanced aerosol-cloud relationships in more stable and adiabatic clouds, Asia-Pacific J. Atmos. Sci., doi:10.1007/s13143-012-0028-0, 2012.

Logan, T., Dong, X. and Xi, B.: Aerosol properties and their impacts on surface CCN at the ARM Southern Great Plains site during the 2011 Midlatitude Continental Convective Clouds Experiment, Adv. Atmos. Sci., doi:10.1007/s00376-017-7033-2, 2018.

McComiskey, A, Feingold, G., Frisch, A. S., Turner, D. D., Miller, M., Chiu, J. C., Min, Q., and Ogren, J.: An assessment of aerosol-cloud interactions in marine stratus clouds based on surface remote sensing, J. Geophys. Res., 114, D09203, doi:10.1029/2008JD011006, 2009.

Parworth, C., Fast, J., Mei, F., Shippert, T., Sivaraman, C., Tilp, A., Watson, T. and Zhang, Q.: Long-term measurements of submicrometer aerosol chemistry at the Southern Great Plains (SGP) using an Aerosol Chemical Speciation Monitor (ACSM), Atmos. Environ., doi:10.1016/j.atmosenv.2015.01.060, 2015.

Wang, Y., Khalizov, A., Levy, M. and Zhang, R.: New Directions: Light absorbing aerosols and their atmospheric impacts, Atmos. Environ., doi:10.1016/j.atmosenv.2013.09.034, 2013.

Wood, R. and Bretherton, C. S.: On the relationship between stratiform low cloud cover and lower-tropospheric stability, J. Clim., doi:10.1175/JCLI3988.1, 2006.

---

## Author Response (ED1)

**Response to Reviewer #1**

We appreciate your time for thoroughly reviewing our manuscript. We would like to thank you for the constructive comments and suggestions that help to improve the manuscript. The manuscript has been revised accordingly. The reviewer's comments are provided in black text and our responses are provided in blue text.

**Response:**

I would like to thank the authors for the different answers to the comments and the changes made in the article. The quality of the paper has increased and the results are now more robust and clear. Nevertheless, some of my comments have not been answered adequately or are not mentioned in the final article. I would like the authors to comment on few points I am referring to here.

From the first general comment answer:
The uncertainty for Nd of 25 % is considered by the author, I find this value excessively low, especially when compared with Grovesnor et al. (2017) values: Their review ranges the uncertainty of Nd from 20 % to 75% from ground based radar measurements, please refer to Section 5 from their paper. The uncertainty of LWP leads to an uncertainty on Nd much greater than 25 % (especially for LWP in the range considered in the study). Also in Dong et al. (1997), they conclude to an uncertainty on Nd of 36 %. I would like to see a deeper analysis on the uncertainty of Nd for the different parameters. Also, I think the paragraph on the uncertainty from the answer should appear in the article and not only the last part about re.

Thanks for the comments.

As reviewed in Grovesnor et al. (2017), Schmidt et al., (2014) used a dual-Field-of-View Raman lidar technique to retrieve $r_e$ and further computed $N_d$ following the equation $N_d = \frac{\alpha}{2\pi l r_e^2}$. They found that the large errors in $r_e$ (25% - 40%) contribute to the $N_d$ relative uncertainty of 50% - 80%. Determination of cloud microphysical properties relies on measurements and calculation methods.

In Dong et al. (1997), the $r_e$ was retrieved based on the method involving spectroradiometer transmittance and LWP. The 10% change in cloud LWP and downward SW at the surface would result in the 10% uncertainty in $r_e$ retrieval.

For the original $N_d$ retrieval in Dong et al. (1997), the uncertainties of cloud droplet concentrations were up to 36%, which were estimated by exerting random perturbations onto the input variables in the retrieval, and then compared the ratios of the standard deviation to the mean. This value of 36% rather represents the sensitivity of $N_d$ to random errors in the retrieval inputs than the uncertainty compared to in-situ measurements.

In this study, the 25% uncertainty of $N_d$ is adapted from the previous experiments compared with the aircraft in situ measurements at the Penn State surface site during the Fall 1996 (Dong et al. 1998) and at the ARM SGP site during March 2000 Cloud Intensive Observational Period (IOP) (Dong et al. 2002; Dong and Mace, 2003). The 25% uncertainty in $N_d$ is statistically estimated based on 5 hours of aircraft in situ measurements during the Fall 1996 and 10 hours of aircraft data during the March 2000 IOP.

Regarding the expression used in this study to compute $N_d$ ($N_d = \left(\frac{3LWP}{4\pi\rho_w r_e^3 \Delta Z}\right) \exp(3\sigma_x^2)$), the uncertainties of input parameters are as follows: 0.15 for logarithmic width of the droplet size distribution (Miles et al., 2000); 20 $gm^{-2}$ for LWP (Liljegren et al. 2001); 60 m for cloud thickness (Widener et al., 2012); and 10% for $r_e$ (Dong et al., 2002).

To assess the contributions of different parameters' uncertainties to $N_d$ retrieval, every input parameter was perturbed by its uncertainty with other parameters held fixed, and then we recalculated $N_d$ for all the samples in this study. The results are presented in the Table below.

| | $N_d$ (cm$^{-3}$) |
|---|---|
| Baseline | 297 |
| $\sigma_x + 0.15$ | 448 (50.8%) |
| $\sigma_x - 0.15$ | 226 (23.9%) |
| LWP $+ 20$ (gm$^{-2}$) | 380 (27.9%) |
| LWP $- 20$ (gm$^{-2}$) | 215 (27.6%) |
| $r_e + 10\%$ (µm) | 224 (24.6%) |
| $r_e - 10\%$ (µm) | 408 (37.4%) |
| $\Delta Z + 60$ (m) | 254 (14.5%) |
| $\Delta Z - 60$ (m) | 366 (23.2%) |

The baseline value is the mean value of $N_d$ from all the samples (same as in Fig. 2). The values in the right column denote the retrieved $N_d$ after perturbing the input variables by their estimated uncertainties. The values in the parentheses are the percentage changes with respect to the baseline value. As shown in the Table, the percentage changes in $N_d$ are due to different uncertainty inputs which range from 14.5% to 50.8%, with the majority falling between 20% and 30%. Note that the largest uncertainty of $N_d$ happens when $\sigma_x$ is increased by 0.15. However, considering that continental stratocumulus generally contain smaller droplets, one might expect their distribution width to be smaller than 0.38 as well (Dong et al., 1997). Therefore, considering the combined contribution of input uncertainties to the $N_d$ retrieval, the overall uncertainty of 25% compared to the aircraft in-situ measurement should be a reasonable estimation. In this case, the mean ratio of $N_d/N_{CCN}$ for the weakly absorptive aerosol regime range from 52% to 86%, while the mean ratio of $N_d/N_{CCN}$ for the strongly absorptive aerosol regime range from 41% ~ 67%.

Accordingly, the statement '*As the results, the 10% change in cloud LWP and downward SW at the surface would result in the 10% uncertainty in $r_e$ retrieval. And the $N_d$ uncertainty is*

*statistically estimated to be 25%, compared with the aircraft in situ measurements at the Penn State surface site during the Fall 1996 (Dong et al. 1998) and at the ARM SGP site during the March 2000 Cloud Intensive Observational Period (IOP) (Dong et al. 2002; Dong and Mace, 2003)*' has been added to Section 2.1.2 in the revised manuscript.

Moreover, the following discussion below has been added to the third paragraph of Section 3.3.4 in the revised manuscript:

'*In addition, the sensitivity and uncertainty of $N_d$ is examined in order to estimate the impact of $N_d$ uncertainty on the assessment of CCN activation rate. To assess the contributions of different input parameter uncertainties to $N_d$ retrieval, every input parameter was perturbed by its own uncertainty with other parameters held fixed. The results are as follows: (a) an increase (decrease) of LWP by 20 $gm^{-2}$ leads to 27.9% (27.6%) change in $N_d$ while an increase (decrease) $\sigma_x$ by 0.15 leads to a 50.8% (23.9%) change in $N_d$; (b) an increase (decrease) cloud thickness by 0.15 leads to a 14.5% (23.2%) change in $N_d$; and (c) an increase (decrease) in $r_e$ by 10% leads to 14.5% (23.2%) change in $N_d$. The percentage changes in $N_d$ due to different input uncertainties range from 14.5% to 50.8%, with the majority falling between 20% and 30%. Note that the largest uncertainty of $N_d$ happens when increasing $\sigma_x$ by 0.15. However, when considering that continental stratocumulus generally contains smaller droplets, one might expect their distribution width to be smaller than 0.38 (Dong et al., 1997). Therefore, the overall uncertainty of 25% compared to the aircraft in-situ measurement should be a reasonable estimation. In this case, the mean ratio of $N_d/N_{CCN}$ for the weakly absorptive aerosol regime range from 52% to 86%, while the mean ratio of $N_d/N_{CCN}$ for the strongly absorptive aerosol regime range from 41% ~ 67%.*'

Also, can you make explicit the formula you use to propagate the uncertainty of re on ACI? With a simple equation, I find an uncertainty on ACI between 0.01 and 0.01 for a change in re of 10% and Nccn of 0 % (optimistic), which is not negligible considering the value from figure 8-b and the conclusion in paragraph 3.3.6. I would like to see some clarifications here.

In the previously revised manuscript, we added 10% of $r_e$ on their original values and deducted 10% of $r_e$ on their original values, then re-do the regression to obtain the upper and lower bounds of the ACIs respectively, which are denoted by the dash lines in the original Fig. 8. The results differ from the original ACIs by only ~0.0003. However, this method treated the $r_e$ uncertainty as uniform and did not consider any different circumstances.

In the newly revised manuscript, we use the Monte Carlo method to propagate the $r_e$ uncertainty on ACI. The procedure is given as follows:
1. For each data point, the $r_e$ value is randomly perturbed to be increased or decreased by 10%.
2. The corresponding ACI is re-calculated based on the perturbed set of $r_e$.
3. Repeat Step 1 & 2 100,000 times.
4. After 100,000 iterations, we obtain a distribution of ACIs
5. The uncertainty of ACI is given by one standard deviation of 100,000 values of ACIs.
Since the distribution of ACIs follows a normal distribution with a narrow peak, this uncertainty value represents the uncertainty in the computed ACI due to error in $r_e$ retrieval.

The uncertainties of ACI for both regimes under the two LWP ranges are denoted in the revised Fig. 8 below:

[Figure]

**Figure 8.** $r_e$ as a function of $N_{CCN}$ and the values of $ACI_r$ under the strongly absorptive (in red) and weakly absorptive (in blue) aerosol regimes at two LWP bins: 0-50 g m$^{-2}$ (a) and 200-250 g m$^{-2}$ (b). Note that the dashed lines denote the uncertainties of $ACI_r$ due to 10 % error in $r_e$ retrieval.

In the lower LWP range (Fig. 8a), the ACI uncertainty is 0.02 (0.03) for weakly (strongly) regime, account for the uncertainties, the difference in ACI between two absorptive regimes is preserved. In the higher LWP range (Fig. 8b), the ACI uncertainty is 0.04 (0.02) for the weakly (strongly) regime, which is non-negligible. Taking the uncertainties of ACI into account, the ACI in the two absorptive regimes cannot be well separated, owing to the enhanced collision-coalescence process accompanied by higher LWP and the diminished cloud response to aerosol associated with different $\omega_{abs}$ values. In general, the 10% uncertainty in $r_e$ retrieval contributes to 0.02 ~ 0.04 in ACI uncertainties.

Accordingly, the following discussion has been added to the second paragraph of Section 3.3.6 in the revised manuscript.

*'In order to assess the impact of $r_e$ uncertainty on $ACI_r$, we use the Monte Carlo method to propagate the $r_e$ uncertainty on $ACI_r$, with the procedure given as follows: For each data point, the $r_e$ value is randomly perturbed to be increased or decreased by 10%, and thus the corresponding $ACI_r$ can be re-calculated based on the perturbed set of $r_e$. After 100,000 iterations, we obtain a distribution of $ACI_r$s. The uncertainty of $ACI_r$ is given by one standard deviation of those 100,000 values of $ACI_r$s, since the distribution of $ACI_r$s follows a normal distribution with a narrow peak, this uncertainty value represents the uncertainty in the computed $ACI_r$ due to errors in the $r_e$ retrieval. The uncertainties of $ACI_r$ for the two absorptive regimes are denoted as the dashed line in Fig. 8. In the lower LWP range (Fig. 8a), the $ACI_r$ uncertainty is 0.020 (0.030) for the weakly (strongly) regime, account for the uncertainties, the difference in $ACI_r$ between the two absorptive regimes is preserved. In the higher LWP range (Fig. 8b), the $ACI_r$ uncertainty is 0.044 (0.023) for the weakly (strongly) regime, which is non-*

*negligible. Taking the uncertainties of ACI$_r$ into account, the ACI$_r$ in the two absorptive regimes cannot be well separated, owing to the enhanced collision-coalescence process accompanied by higher LWP and the diminished cloud response to aerosols associated with different $\omega_{abs}$ values. In general, the 10% uncertainty in r$_e$ retrieval contributes to 0.02 ~ 0.04 in ACI$_r$ uncertainties.'*

From the second general comment answer:
In the article (p. 17, l. 3 of the new document), the lack of vertical velocity measurements is mentioned. But, it can be retrieved from ECMWF (with LTS), it would not be directly at cloud base and will concern a large-scale value but did you look at this parameter? I would also be curious to see the effect of humidity on the results. Did you look at other meteorological parameters than LTS which can impact the ACI?

Thanks for the comments.

Yes, we have checked the vertical velocity in pressure coordinate (Omega) values at the 925 hPa level for all sample points, which represent the large-scale forcing on the vertical motion between surface and cloud-layer over the SGP site. The results are shown in Fig. R1 below:

[Figure]

Figure R1

As shown in Fig. R1a, both absorptive regimes shared the same mean value of 0.031 Pa/s with a slightly different distribution. In addition, the majority of Omega values in both regimes are greater than 0, which means the large-scale environments over the SGP are generally dominated by sinking motion. Fig. R1b shows the Omega values sorted by LWP. The vertical velocities are not correlated with $\omega_{abs}$ and show no dependence on LWP.

Given that all the cases happened during the Spring and Winter seasons, the distributions of Omega are greatly influenced by large-scale synoptic patterns. To examine the synoptic patterns associated with them, we obtained the geopotential height data from ERA5 reanalysis and applied the composite analysis on the cases that in weakly/strongly absorptive regime. Results for 925 hPa level are shown in the figure below:

[Figure]

Figure R2

Fig. R2 (a) 925 hPa composite geopotential height in the weakly absorptive regime with the SGP site located ahead of the ridge. (b) 925 hPa composite in the strongly absorptive regime with the SGP site within a large domain of relatively high pressure. As for higher pressure level up to 700 hPa (Fig. R2c & d), the SGP site located ahead of the ridge for both regimes. Therefore, the synoptic patterns are favorable for generation of downward motion at the lower troposphere, and the sinking motion induces relatively stable environments in the lower troposphere which is consistent with the LTS measurements.

Since the Omega value is obtained from a relatively larger domain surrounding SGP, it is hard to reflect the true cloud-scale dynamics, especially the vertical velocity or turbulence strength at the cloud base.

Accordingly, the following discussion has been added to the fifth paragraph of Section 3.3.4 in the revised manuscript.

*'In addition, the vertical velocity in pressure coordinate (Omega) values at the 925 hPa level, which represent the large-scale forcing on the vertical motion between surface and cloud-layer, are also sorted by LWP for the two absorptive regimes in order to check the potential influence of the environmental dynamic state (figure not shown). However, the Omega for both absorptive regimes share the same mean value of 0.031 Pa/s and show no dependence on LWP, indicate that the large-scale environments over the SGP are generally dominated by sinking*

*motion. The synoptic patterns of composite geopotential height for the two absorptive regimes show that the SGP site is located ahead of the 700 hPa ridge and is located within the 925 hPa high. The meteorological pattern is favorable for generation of downward motion at the lower troposphere, and the sinking motion induces relatively stable environments in the lower troposphere which is consistent with the LTS measurements. Considering the fact that the Omega value is obtained from a relatively larger domain surrounding the SGP, it is difficult to reflect the true cloud-scale dynamics, especially the vertical velocity or turbulence strength at the cloud base. Therefore, the influence of cloud-scale dynamics, presumably cloud-base updraft, is not negligible since the sensitivity of cloud droplet to aerosol loading is enhanced with increasing updraft velocity as reported in previous studies (e.g., Feingold et al., 2003; McComiskey et al., 2009).'*

Additionally, we also checked the relative humidity (RH) in both regimes to examine whether there is any causality between RH and ACI. The RH is obtained from the Merged-Sounding product at SGP, then calculated as the mean value within the cloud-layer at every sample point, the RH distributions and dependences on LWP in two regimes are presented in the figure below:

[Figure]

Figure R3

As shown in Fig. R3(a), the cloud-layer mean RH in the weakly absorptive regime share a similar distribution as in the strongly absorptive regime, with a 1.6% difference in the mean value. More samples with relatively lower RH were found in the strongly absorptive regime. When the RH data are sorted by LWP (Fig. R3b), the values in the weakly absorptive regime are also higher than those in the strongly absorptive regime. This discrepancy is in accord with the discussion of the absorptive effect of aerosol on the environment, with the presence of strongly light-absorbing aerosols, the cloud layer heating induced by the aerosol absorptive effect can result in the reduction of in-cloud relative humidity (Wang et al., 2013).

Accordingly, the statement *'Furthermore, the radiative effect of light-absorbing aerosols on the cloud environment cannot be neglected, since the strongly light-absorbing aerosols can absorb solar radiation and heat the in-cloud atmosphere by emission, which results in the reduction of relative humidity (or supersaturation) in the cloud layer (Bond et al., 2013; Wang et al., 2013). This effect is evident by the observation as the values of in-cloud relative humidity*

*in the strongly absorptive regime are slightly lower than those in the weakly absorptive regime.'* has been added to the sixth paragraph of Section 3.3.4 in the revised manuscript.

The stratus formation is enhanced by high LTS (Klein et al. 1993) and, therefore, more prone to high ACI, which is in line with the results presented. The results show that ACI is a function of LTS and a function of wabs. Moreover, from the Figures (a) and (b) from the answer of the second general comment, LTS and wabs seem correlated with each other. The effect of LTS on ACI does not seem negligible, maybe even larger than the effect of wabs on ACI. I think a discussion is needed about the conclusions of the article and the correlations from observation that are not necessarily causality, and that the observed effect can be inhibited or enhanced (Grysperdt, 2016). You mention it but I think it would help to see the ACI for the two regimes of wabs and constrained for two regimes of LTS to ensure that the results described by Figure 8 are not an artifact.

Thanks for the comments.

In order to constrain the ACI for the two regimes by their LTS values, we adapted the criteria described in Grysperdt et al. (2016) that the LTS value of 18K denotes the demarcation line between high and low LTS regimes. However, the sample points that fall into the low LTS category are highly limited, largely owing to the selected cases. In 0 - 50 $\mathrm{gm^{-2}}$ LWP range, only 3(6) samples in weakly (strongly) regime have LTS values that are lower than 18K; in $200-250$ $\mathrm{gm^{-2}}$ LWP range, only 11 (12) samples in weakly (strongly) regime are lower than 18K. Since the calculation constrained in the low LTS regime cannot pass the 95% significant test, the figure below shows the ACI constrained in high LTS regime only.

[Figure]

Figure R4

In 0 - 50 $\mathrm{gm^{-2}}$ LWP range (Fig. R4a), after considering only the sample points with LTS higher than 18K, the mean values of LTS in the two regimes are closer. The ACI for the weakly absorptive regime is enhanced from 0.26 to 0.31, and ACI for the strongly absorptive regime

is increased from 0.21 to 0.24. The enhancement effect of LTS on the ACI is noticeable, which is in accordance with the previous discussion that a high LTS environment is associated with sufficient turbulence in the boundary layer and a closer connection between surface and cloud layer, and thus enhance the cloud microphysical responses to the CCN below. Note that though ACIs are increased for both regimes, the difference between them becomes larger. Owing to the low-level stratus is more susceptible to weakly absorptive aerosol, the enhancement of ACI induced under a high LTS environment is better manifested in the weakly absorptive regime. As for $200 - 250$ $gm^{-2}$ LWP range (Fig. R4b), the LTS effects on ACI are less significant compared to low LWP range. No significant change in weakly absorptive regime and the ACI in strongly absorptive regime decreased from 0.12 to 0.10, partly owing to the enhanced collision-coalescence process accompanied by higher LWP, and thus inhibit the impact of LTS on ACI.

Overall, ACIs are enhanced under the high LTS, but the difference between two regimes do exist, indicating that both $\omega_{abs}$ and LTS can be the impact factor of the ACI but they are not necessarily having causality between them.

Accordingly, the following discussion has been added to the third paragraph of Section 3.3.6 in the revised manuscript.
*'In order to quantify the impact of LTS on $ACI_r$, we adapted the criteria described in Grysperdt et al. (2016) that the LTS value of 18K denotes the demarcation line between high and low LTS regimes, and constrain the $ACI_r$ for the two regimes by their LTS values accordingly. Owing to the highly limited sample points that fall into the low LTS category, the $ACI_c$ can only be constrained in the high LTS condition. For the $0 - 50$ $gm^{-2}$ LWP range, the $ACI_r$ for the weakly absorptive regime increases from 0.26 to 0.31, and the $ACI_r$ for the strongly absorptive regime increases from 0.21 to 0.24. The enhancement effect of LTS on the $ACI_r$ is noticeable, which in accordance with the previous discussion that high LTS environment is associated with (a) sufficient turbulence in the boundary layer and (b) a closer connection between the surface and cloud layer, which enhances the cloud microphysical responses to the CCN. Note that though $ACI_r$s are increased for both regimes, the difference between them becomes larger (from 0.05 to 0.07) because low-level stratus clouds are more susceptible to weakly absorptive aerosol. Furthermore, the enhancement of $ACI_r$ in the high LTS environment is more evident in the weakly absorptive regime. In the case of the $200 - 250$ $gm^{-2}$ LWP range, the LTS effects on $ACI_r$ are less significant compared to the lower LWP range. No significant change in weakly absorptive regime is evident and the $ACI_r$ in the strongly absorptive regime decreased from 0.12 to 0.10, partly owing to the enhanced collision-coalescence process accompanied by higher LWP, and thus inhibits the impact of LTS on $ACI_r$. Overall, $ACI_r$s are enhanced under high LTS conditions, but the difference between two regimes indicates that both $\omega_{abs}$ and LTS can be the impact factor of the $ACI_r$ but they are not necessarily having causality between them.'*

From specific comment:

I suggest to put the location SGP at the end of the second sentence: "… are selected over the Southern Great Plains region of the United States (SGP). The physicochemical…" Otherwise, the logic of the abstract is difficult to follow.

Thanks for the suggestion.

The sentence has been changed to '*a total of 16 low-level stratus cloud cases under daytime coupled boundary layer conditions are selected over the Southern Great Plains region of the United States (SGP)*' in the revised manuscript.

The answers of the following comment should appear in the text:
- the different resolution and uncertainty between KAZR and MMCR

The following statement '*Although the uncertainty of KAZR (~30m) is lower than MMCR (~45m), the difference of 15m between these two cloud radars would not cause a significant difference in detecting the cloud boundaries*' has been added to section 2.1.1 in the revised manuscript.

- The comparison with aircraft measurements from Delle Monache et al. (2004) with a quantification of their results to assess the reliability of the measurements.

The following statement '*A study was conducted by Delle Monache et al., (2004) used in-situ aerosol measurements from 59 flights from March 2000 – March 2001 to compare with the surface aerosol measurements. Their results showed that the aerosol extensive properties such as the total extinction by particles measured within the well-mixed boundary layer were well-correlated with surface measurements with the $R^2$ value of 0.88.* ' has been added to section 2.3 in the revised manuscript.

- The threshold you are using are from Dong et al. (2015), originally suggested by Jones et al. (2011)

The following statement '*These thresholds are the same as in Dong et al. (2015), originally suggested by Jones et al. (2011)*' has been added to section 2.3 in the revised manuscript.

- The uncertainty of ACI corresponds to the 95 % confidence interval.

The following statement '*The $ACI_r$ values range from 0.09 – 0.24 with a mean value of 0.145 ± 0.05, the uncertainty of $ACI_r$ corresponds to the 95 % confidence interval.*' has been added to section 3.2 in the revised manuscript.

**Response to Reviewer #2**

We appreciate your time for thoroughly reviewing our manuscript. We would like to thank you for the constructive comments and suggestions that help to improve the manuscript. The manuscript has been revised accordingly. The reviewer's comments are provided in black text and our responses are provided in blue text.

**Response:**

The authors have revised the manuscript in meaningful ways in response to the first reviews but there are still some passages that give a mixed impression of their understanding of the concerns raised in the reviews. The comments here fall somewhere between suggesting minor and major revisions.

One of my reservations about the first version of this manuscript was the way in which processes for cloud drop (Nd) activation were described relative to measurements of total aerosol number concentrations (Na) and cloud condensation nuclei (CCN.) The analysis of the observations showing the difference in Na conversion to CCN in weakly and strongly absorptive regimes is interesting and fairly straightforward (with the exception noted in the paragraph below.) This analysis is done for an observed %S of 0.2. While there is some discussion of how different %S might affect the results it should be clearly stated in the conclusions that they rest on this chosen value of %S.

Thanks for the comments.

The following discussion about how different %S might affect the results has been added to the first paragraph of Section 3.3.3 in the revised manuscript:

*'Note that those ratios are computed for an observed supersaturation level of 0.2%. The fraction of aerosols that can activate as CCN increases with an increase in supersaturation level, under the same aerosol size and composition condition (Dusek et al., 2006). A sensitivity test of how the aerosol activation rate varies with different supersaturation levels is done by first interpolating the $N_{CCN}$ from 0.2 % to 1.15 % and then calculating the $N_{CCN}/N_a$. As a result, the ratios of $N_{CCN}/N_a$ for the weakly absorptive regime range from 0.54 to 0.38, while the ratios for the strongly absorptive regime range from 0.45 to 0.25. Considering that the supersaturation level in the continental boundary layer stratus is nearly impossible to reach the level of 1.15%, the supersaturation level of 0.2% used in this study, which represents the most typical condition for continental low stratus, yields reasonable results.'*

With respect to the effect of aerosol composition/absorption on the efficacy of particles to serve as CCN, on P13 L23-25 you interpret the data from Fig 6. You note that there are ranges of Na for which the more absorbing aerosol are below the weakly absorbing aerosol. Are the ranges of number concentration meaningful in some way? What I see are two slopes that are very much alike except for defined excursions that you call out in these ranges. What's happening for those particular observations? What makes them different? It raises the question of why some highly absorbing aerosol fall within the same relationship of the weakly absorbing aerosol

and some do not. Is there another or additional property of this latter set of aerosol has that make them poor CCN?

Thanks for the comments.

The different relationships between the two absorptive regimes shown in Fig 6 are the result of the mixed effect of $\omega_{abs}$, water availability and aerosol size on the aerosol activation.

Looking closely to Fig 6, we defined three clusters of different $N_a$ range: 200-500 $cm^{-3}$, 500 – 1100 $cm^{-3}$ and 1100-3500 $cm^{-3}$. The mean values of LWP, AE and ratio of $N_{CCN}$ to $N_a$ in each range are shown in the Table below:

| $N_a$ Range (cm$^{-3}$) | LWP (gm$^{-2}$) | | AE | | $N_{CCN}/N_a$ | |
|---|---|---|---|---|---|---|
| | Weakly | Strongly | Weakly | Strongly | Weakly | Strongly |
| (a)200-500 | 158 | 162 | 1.59 | 1.73 | 0.77 | 0.35 |
| (b)500-1100 | 138 | 167 | 1.53 | 1.40 | 0.58 | 0.51 |
| (c)1100-3500 | 95 | 127 | 1.67 | 1.57 | 0.42 | 0.32 |

In 200-500 $cm^{-3}$ $N_a$ range, the samples from the two absorptive regimes are the most separated. The mean values of LWP indicate relatively sufficient water availability with less aerosol concentration, accompanied by the fact that the weakly absorbing aerosol sizes are relatively larger indicated by smaller AE. It is known that larger aerosol particles easily activate under same composition (Dusek et al., 2006), considering the weakly absorbing aerosol is more hydrophilic, the largest conversion rate difference among these three ranges are to be expected.

In 500-1100 $cm^{-3}$ $N_a$ range, the AE (LWP) value for strongly absorbing aerosols is noticeably smaller (higher) than those in the weakly absorptive regime. Therefore, the combined effect of more large particles and more water in the strongly absorptive regime lead to a $N_{CCN}/N_a$ ratio close to the $N_{CCN}/N_a$ ratio in the weakly absorptive regime.

In 1100-3500 $cm^{-3}$ $N_a$ range, there are more smaller aerosol particles and less water availability which leads to a greater aerosol competing effect with respect to water among both absorptive regimes, resulting in the lowest activation rate among the three ranges in both regimes.

Accordingly, the discussion above has been added to the third paragraph of Section 3.3.3 in the revised manuscript.

Therefore, to better examine the difference in aerosol activation capacity, Fig. 7 shows the $N_{CCN}/N_a$ sorted by LWP for the two absorptive regimes. Moreover, the AE values are also sorted by LWP, thereby ruling out the influence of LWP and AE on aerosol activation. As shown in the Figure below, in every LWP bin, the AE for the weakly absorptive regime is either higher than or very close to the AE for the strongly absorptive regime. Even with relatively smaller particle sizes, under similar water availability, the weakly absorbing aerosol

can better activate as CCN. In conclusion, the significant effect of aerosol composition, which is inferred by aerosol absorbing ability, on the aerosol activation capacity does exist.

[Figure]

Accordingly, the following discussion has been added to the fourth paragraph of Section 3.3.3 in the revised manuscript.

'*As shown in Fig. 6, for three $N_a$ ranges (200 - 500; 500 - 1100 and 1100 - 3500 $cm^{-3}$), the strongly absorbing aerosols show different relationships compared to weakly absorbing aerosols. The mean $N_{CCN}/N_a$ values for those three $N_a$ ranges for weakly absorptive regimes are 0.77, 0.58, and 0.42, respectively, while the mean $N_{CCN}/N_a$ values for the strongly absorptive regimes are 0.35, 0.51, and 0.32, respectively. This phenomenon is due to the mixed effect of aerosol composition (inferred by absorbing ability), aerosol size, and water availability on the aerosol activation. In the 200 - 500 $cm^{-3}$ $N_a$ range, where the samples from the two absorptive regimes are most separated. The mean values of LWP (158 $gm^{-2}$/162 $gm^{-2}$ for weakly/strongly absorptive regimes) indicate relatively sufficient water availability with less aerosol concentration. In addition, the weakly absorbing aerosol sizes are larger (AE = 1.59) than the strongly absorbing aerosol (AE = 1.73). It is known that larger aerosol particles easily activate under same composition (Dusek et al., 2006), considering the weakly absorbing aerosol is more hydrophilic, the largest conversion rate difference among these three ranges are to be expected. The samples in the 500 - 1100 $cm^{-3}$ $N_a$ range, have AE value (1.40) for strongly absorbing aerosols are noticeably smaller than those in the weakly absorptive regime (1.53), and the LWP in strongly absorptive regime (167 $gm^{-2}$) is much higher than in weakly absorptive regime (138 $gm^{-2}$). Therefore, the combined effect of larger particles and more water in strongly absorptive regime lead to the $N_{CCN}/N_a$ is close to the $N_{CCN}/N_a$ ratio in weakly absorptive regime. The samples in the 1100 - 3500 $cm^{-3}$ $N_a$ range exhibit smaller (AE = 1.67/1.57 for weakly/strongly absorptive regimes) aerosol particle size and less water availability (LWP = 95 $gm^{-2}$/127 $gm^{-2}$ for weakly/strongly absorptive regimes) which results in the lowest activation rate ($N_{CCN}/N_a$ ratio = 0.42/0.32 for weakly/strongly absorptive regimes) among the three ranges for both regimes.*'

The statement '*Furthermore, in the following section, the values of $N_{CCN}/N_a$ and AE are sorted by LWP for the two absorptive regimes, in order to rule out the influence of LWP and AE on aerosol activation to the utmost extent*' has been added to last paragraph of Section 3.3.3 in the revised manuscript.

The statement *'Moreover, in every LWP bin, the AE value for the weakly absorptive regime is either higher than or very close to the AE value for the strongly absorptive regime (Figure not shown). Even with relatively smaller particle sizes, under similar water availability, the weakly absorbing aerosol can better activate as CCN. In conclusion, the significant effect of aerosol composition, which inferred by aerosol absorbing ability, on the aerosol activation capacity does exist*' has been added to second paragraph of Section 3.3.4 in the revised manuscript.

In the previous version of the manuscript there was not a clear message that the conversion of Na to CCN was dependent on aerosol properties and the conversion of CCN to Nd primarily dependent on cloud state and dynamics. In the discussions in Section 3.3. and later this has been largely rectified. The authors provide an explanation that the CCN to Nd relationship is impacted by environmental heating of absorbing aerosol and the resulting effects on cloud microphysics (rather than differences in cloud microphysics being due to the activation process associated with strongly and weekly absorbing aerosol.) The details of this dynamic are not explored due to a lack of relevant observation.

Given that the latter is not an aerosol-cloud microphysical effect there are passages earlier in the paper that are still confusing. On P2 L5-8 the implication is still that there is some microphysical effect for CCN to Nd that is different for absorbing and non-absorbing aerosol.

We have revised this statement to emphasize that the difference in CCN to $N_d$ conversions is the result of different thermodynamic states and the environmental heating effect of strongly absorbing aerosol.

Accordingly, the corresponding part of the abstract has been changed to '*In terms of the sensitivity of cloud droplet number concentration (Nd) to CCN, the conversion ratio of $N_d/N_{CCN}$ for weakly (strongly) absorptive aerosols is 0.69 (0.54), owing to the combined effect of different cloud dynamic, thermodynamic states and cloud-layer heating effect of the strongly light-absorbing aerosol. The measured $ACI_r$ values in the weakly absorptive regime are relatively higher, indicating that clouds have greater microphysical responses to aerosols in weakly absorptive regime than in strongly absorptive regime. Consequently, we expect larger shortwave radiative cooling effect from clouds in the weakly absorptive regime than those in the strongly absorptive regime.*'

Some of the problem may just be the use of terminology. For Example, on P3 L1 "the efficacy of the activation of CCN" would be clearer as "the efficacy of the activation of aerosol." For the CCN measurements as they are made here, at a given supersaturation, the number concentration is assumed to be 100% efficacious for cloud droplet activation. Size and composition, however, will affect the activation rate of aerosol.

Thanks for the suggestion.

The sentence has been changed to '*The efficacy of the activation of aerosol has been widely known to be influenced by aerosol size distribution and chemical composition which are the primary sources of uncertainty in assessing ACI*' in the revised manuscript.

On P10 L28-29 to P11 first paragraph there is a statement that Na does not serve as a realistic CCN proxy for calculating ACIr but attributes the poor result on decoupled boundary layer condition. Is this the author's assumption or the assumption made by Feingold et al. 2003? What about the fact that Na can include large numbers of very small particles that will not activate, especially at these supersaturations? The difference between Na and scattering as a CCN proxy and coupling of the BL are mixed up in this discussion. Scattering (or aerosol index) have long been used as a more reliable CCN proxy due to their sensitivity to larger particle sizes. It just contributes to the confusion with how the manuscript characterizes total aerosol and CCN with respect to droplet activation.

Thanks for the comments.

The statements in P10 L28-29 to P11 L5 are the assumptions and findings in Feingold et al. (2006). To avoid further confusion, the discussion about the role of boundary layer coupling is omitted in the revised paragraph, since the usefulness of using surface aerosol measurement to represent aerosol aloft under coupled boundary layer condition has been discussed in Section 2.3.

Therefore, the discussion in Section 3.2 of the revised manuscript has been changed as follows: '*Previous studies have focused on the aerosol-cloud interaction in stratocumulus clouds at the ARM SGP site. Based on the analysis of seven selected stratocumulus cases during the period 1998 - 2000, Feingold et al. (2003) reported the first ground-based measured ACI$_r$ values of 0.02 to 0.16 using the lidar measured aerosol extinction at a wavelength of 355 nm as the proxy for aerosol loading. A later study conducted by Feingold et al. (2006) assessed the ACI$_r$ using different aerosol measurements as CCN proxies, in three selected stratus cases during the intensive operation period in May 2003. They found that the ACI$_r$ values were unrealistic when using N$_a$ to represent CCN loading while using the surface aerosol scattering coefficient (σ$_{sp}$) and aerosol extinction at an altitude of 350 m as CCN proxies yield similar ACI$_r$ values ranging from 0.14 to 0.39 (Feingold et al. 2006). The assessment of ACI$_r$ can be largely affected by the usage of different aerosol measurements that served as CCN proxies due to their own characteristics. Aerosol scattering and extinction coefficient are known to be relatively reliable CCN proxies since they are more sensitive to aerosols that have larger particle sizes. As for N$_a$, which represents the concentration of aerosol particles with diameters larger than 10 nm, it is likely to pick up the very small aerosols generated by new particle formation events. This proportion of aerosols is presumably hard to activate as CCN so that would not be counted in N$_{CCN}$, especially under the 0.2% supersaturation used in this study. Hence, it is less representative to use N$_a$ to accurately represent N$_{CCN}$ without the prior knowledge of the aerosol capacity to activate as CCN. Therefore, the usage of N$_{CCN}$ in this study is favorable to*

*yield a more straightforward assessment of ACI$_r$, since the CCN measurement directly represents the amount of aerosol droplets that already activated and have the potential of further growth.*'

This continues in the next paragraph, P11 L10. Is there an assumption that a constant fraction of aerosol effective activates as you state? At what %S? You've shown yourself this relationship can depend on composition and it certainly will depend on size and that is known to be variable at SGP. New particle formation events can push aerosol concentrations up with a large number concentration at very small sizes that you would not see in the CCN or scattering measurements. I don't think the paragraph is the right way to lead into your results.

Thanks for the comments.

The discussion in this paragraph was meant to suggest that the statement 'a nearly constant fraction of aerosol effectively activates as CCN' and 'aerosol loading is more important than the aerosol size and composition' are not true in this study, and therefore the role of aerosol species is necessary to be examined. We agree that the way of presentation here conveys a confusing message. In order to avoid further confusion, this part of the discussion has been changed as follows:

'*In order to better understand the aerosol particle activation process in typical continental low-level stratus clouds, in the latter part of this study, the ratios between $N_{CCN}$ and $N_a$ are examined and used to represent the aerosol activating capacities. Since the aerosol activating capacities greatly depend on the aerosol sizes and compositions, in order to further examine the role of aerosol species in aerosol activation process and the potential impact on ACI$_r$, the samples from the 16 selected cases are divided into two groups according to their absorptive regime which is discussed in the following section.*'

Ultimately I would suggest leaving the Na measurements out of the discussion when covering CCN to Nd or ACI. Na should be the focus when discussing how absorption impacts how well aerosol serve as CCN. Because you have CCN measurements and can use these directly to reference ACI in terms of drop numbers or effective radius just leave it at that. You can still go back and show that ACI differs in high and low absorption regimes by using the co-SSA quantity and leave total number concentration out of it. I don't see the need for referring to a CCN proxy when you have CCN. It's really confusing the discussion and leaves the reader wondering how the authors are considering the cloud drop activation process. Related, P20 L29-P21 L1 - not sure this statement is worth making.

Thanks for the suggestions.

The relevant discussions have been modified accordingly in the revised manuscript.

The revised Fig. 8 below now showing only the ACI$_r$ with respect to CCN.

[Figure]

**Figure 8.** $r_e$ as a function of $N_{CCN}$ and the values of $ACI_r$ under the strongly absorptive (in red) and weakly absorptive (in blue) aerosol regimes at two LWP bins: 0-50 g m$^{-2}$ (a) and 200-250 g m$^{-2}$ (b). Note that the dashed lines denote the uncertainties of $ACI_r$ due to 10 % error in $r_e$ retrieval.

The statement at P20 L29-P21 L1 has been deleted, and the last two paragraphs of Section 4 has been changed as follow:

*'The conversion rates of $N_d/N_{CCN}$ for the weakly absorbing aerosols (mean ratio of 0.67) are higher than for the strongly absorbing aerosols (mean ratio of 0.54). Partly owing to the higher LTS environment for the weakly absorptive regime which enhance the connection between cloud and the below-cloud moisture and CCN. Also the cloud layer heating effect induced by the strongly light-absorbing aerosols results in the reduction of in-cloud supersaturation and leads to the damping of CCN activation process for the strongly absorptive regime. As a result, cloud droplets that form from weakly absorbing aerosols tend to have smaller sizes and higher concentrations than cloud droplets forming from strongly absorbing aerosols. Furthermore, the cloud droplets in the weakly absorptive regime exhibit a greater growing ability, as given by larger $r_e$ values that increase with LWP under similar $N_d$. The differences in cloud droplet development between the two regimes is a likely result of the combination of thermodynamics, dynamics, and aerosol radiative effects.*

*Under low LWP conditions, the measured $ACI_r$ values in the weakly absorptive regime are relatively higher, indicating that clouds have greater microphysical responses to aerosols in weakly absorptive regime than in strongly absorptive regime, owing to favorable LTS condition in the weakly absorptive regime, and the cloud-layer heating effect of light-absorbing aerosol in the strongly absorptive regime. The observed $ACI_r$ is enhanced after constrained by high LTS. Under higher LWP conditions, the enhanced collision-coalescence process diminish the LTS impact on $ACI_r$, and the damping of $ACI_r$ is more evident, which is consistent with the results from all the cases. In general, the 10% uncertainty in $r_e$ retrieval contribute to $ACI_r$ uncertainties range from 0.02 to 0.04 for the two absorptive regime, with the $ACI_r$ difference between the two absorptive regimes still well-preserved. As a result, clouds that develop from*

*weakly absorbing aerosols exhibit a stronger shortwave cloud radiative effect than clouds originating from strongly absorbing aerosols. Additional future work will focus on investigating detail composition of different aerosol plumes, with respect to their physicochemical properties. The aerosol-cloud-interaction processes under the influence of different aerosol types associated with airmasses and the sensitivity to dynamic and thermodynamic factors will be further examined.'*

In addition, some discussion of the heating effect of absorbing aerosol on the environment leading to differences in stability and cloud state/dynamics (updraft speeds and moisture availability) early in the manuscript might help to clarify the different effects of aerosol absorption on the cloud properties that you are postulating.

Thanks for the suggestion.

The further discussion of the heating effect of absorbing aerosol on the environment leading to differences in stability and cloud state/dynamics has been added to the last paragraph of Section 3.3.4 in the revised manuscript, which is as follows:

'*Furthermore, the radiative effect of light-absorbing aerosols on the cloud environment also cannot be neglected, since the strongly light-absorbing aerosols can absorb solar radiation and heat the in-cloud atmosphere by emission, which results in the reduction of relative humidity or supersaturation in the cloud layer (Bond et al., 2013; Wang et al., 2013). Additionally, this aerosol heating effect disrupts the boundary layer temperature structure by enhanced warming aloft, and consequently, inhibits the vertical transport of sensible and latent heat between surface and cloud layer. The impacts of light-absorbing aerosol on cloud-scale thermodynamics and dynamics state might eventually dampen the conversion process from CCN to cloud droplet.'*

I think this manuscript still need work before being considered for publication.

Thanks again for the thorough review of this manuscript.

Specific:
P1 L 26: I think this is supposed to read "…suggests a diminished cloud microphysical response to aerosol loading"

The sentence has been changed to 'The magnitude of $ACI_r$ decreases with increasing LWP which suggests a diminished cloud microphysical response to aerosol loading presumably due to enhanced collision-coalescence processes and enlarged particle size.'.

P6 L8: Aerosol Observing System

The correction has been made.

P7 L5 "question of whether surface aerosols can be linked to what actually happens in clouds aloft."

The correction has been made.

P15 L15 – switch position of Nccn and Na

The correction has been made.

**References**

[revised manuscript text omitted]

---

## Author Response (AR3)

**Response to Editor comments**

Dear Dr. Feingold, thank you for the constructive comments and suggestions on the manuscript. They are greatly appreciated. We have carefully studied your comments and revised the manuscript accordingly. In the response below, the reviewer's comments are provided in black text and our responses are provided in blue text.

**Response:**

**Main points:**

Just how much absorbing aerosol is there in the high aerosol absorption events? Can you provide the heating rates? Would they really affect the cloud absorption or LTS?
 P2 L13, General question: you sort your data by single scattering albedo but a low vale of SSA doesn't necessarily mean strong heating unless you have enough aerosol to begin with.

**Thanks for the comments.**

The aerosol number concentration in strongly absorptive regime range from  $150 - 3200 \text{ cm}^{-3}$ , with a mean value of 948 cm-3 and a standard deviation of 667 cm-3. Most events happened under moderate to polluted conditions.

Getting an accurate heating rate profile requires running a radiative transfer model with the input of observed aerosol vertical profiles and atmospheric states, which are unavailable in the present project.

A previous modeling study conducted at the ARM SGP site by Lin et al. (2016) estimated the shortwave heating rates in cloud layers by contrasting the simulations with and without lightabsorbing aerosols. The inclusion of light-absorbing aerosols was represented by an internal aerosol mixture with a mass combination of 95% ammonium sulfate and 5% black carbon. The SSA of this mixture is calculated to be roughly 0.9, as documented in the previous study of Wang et al. (2014). The different values of SSA used in their study (0.9 for light-absorbing and 1.0 for non-absorbing) are comparable to this study (0.89 for strongly-absorbing and 0.97 for weakly-absorbing). The induced increments in cloud-layer shortwave heating rates have a maximum value of 3 K/day, compared to the simulation without light-absorbing aerosols. Note that the aerosol number concentration in Lin et al. (2016) was set to 2800  $\text{ cm}^{-3}$ . To get a simple comparison with the aerosol number concentration in this study, one might expect the light-absorbing aerosols induced cloud-layer shortwave heating rates can have a similar maximum increment, and the general increment should be about 1 K/day which is nonnegligible. The absorption of solar radiation by light-absorbing aerosols warm the cloud layer as well as the boundary layer below it, which in turn, stabilizes the lower troposphere and results in reduced cloud susceptibility.

The discussion above has been added to the fourth paragraph of section 3.3.6 in the revised manuscript.

As for the LTS in the strongly absorptive regime, since it captured larger-scale thermodynamic conditions, mainly impacted by the potential temperature profile between boundary layer top and 700 hPa. While the heating effect induced by the strong absorbing aerosol would cause local perturbation to the temperature structure on a smaller scale. Therefore, we believe that the LTS might be more impacted by the synoptic pattern during the event.

In the future, we would like to focus on the examination of long-term light-absorbing aerosol induced shortwave heating rate at the SGP site using observation data, particularly with adequate methodology.

2) The message seems to me unclear. Is the absorbing aerosol reducing ACI because of microphysics (aerosol composition), or because of stronger absorption / association with weaker LTS? This message should be absolutely clear and consistent in both abstract and conclusions.

P2 L14, I really feel you need to clarify whether the ACI changes are microphysical, or 'indirect', via association with LTS or absorption. The changes leave me confused because they are add-ons rather than integrated changes.

P27 L6, So this is not a microphysical effect of the aerosol, but rather an effect associated with LTS and absorption. This is an important point!

Thanks for the comments.

The difference in the aerosol activation process  $(N_a/N_{CCN})$  between the two regimes is impacted by the microphysics effect of aerosols, particularly the aerosol compositions inferred by the aerosol absorptive properties. While the differences in the  $N_d/N_{CCN}$  and the ACIr between the two regimes are mainly due to the thermodynamic effects associated with aerosol heating and LTS.

Accordingly, the latter part of the abstract has been changed as follows:

'...Furthermore, the mean activation ratio of aerosols to CCN ( $N_{CCN}/N_a$ ) for weakly (strongly) absorbing aerosols is 0.54 (0.45), owing to the aerosol microphysical effects, particularly the different aerosol compositions inferred by their absorptive properties. In terms of the sensitivity of cloud droplet number concentration ( $N_d$ ) to  $N_{CCN}$ , the fraction of CCN that converted to cloud droplets ( $N_d/N_{CCN}$ ) for the weakly (strongly) absorptive regime is 0.69 (0.54). The measured ACIr values in the weakly absorptive regime are relatively higher, indicating that clouds have greater microphysical responses to aerosols, owing to the favorable thermodynamic condition. The reduced ACIrs in the strongly absorptive regime are due to the cloud-layer heating effect induced by strong light-absorbing aerosols. Consequently, we expect larger shortwave radiative cooling effects from clouds in the weakly absorptive regime than those in the strongly absorptive regime.'

These different aerosol effects on different activation processes have been separately discussed and clarified in the rest of the revised manuscript.

3) I'm confused about why collision-coalescence reduces ACIr. I'm not convinced McComiskey et al. (2009) got it right. I am happy to be convinced otherwise.

P11 L18, Considering re vs aerosol, if Nd is decreased because of collision-coalescence, then r\_e will be larger, particularly at smaller aerosol concentrations, which will increase the ACIr. I'm not sure the arguments in McComiskey (2009) were correct. Please give this some thought. P23 L25, Can you clarify your thinking about how collision-coalescence affects ACI? I'm not convinced that McComiskey et al. 2012 got it right.

**Thanks for the comment.**

After a deeper thought, we believe that the more plausible explanation is about condensational growth processes, rather than the collision-coalescence processes.

Given that the ACIr describes the response of  $r_e$  to  $N_{CCN}$  change, under low LWP conditions with more CCN entering the cloud, the smaller particles compete against each other for the limited water supply and cannot efficiently grow into larger sizes. In that case, the higher CCN loading could result in smaller  $r_e$ , and thus the variable range of  $r_e$  is relatively broad, which is reflected by enhanced ACIr. Under high LWP conditions typically associated with sufficient water supply, the newly activated cloud droplet can grow larger quickly via condensation. However, the efficacy of condensational growth decreases with enlarged particle size. The enhanced condensational growth under high LWP conditions can shift the cloud droplet population to larger sizes. Therefore, for a similar CCN perturbation, the variable range of  $r_e$  is narrower, which is reflected by reduced ACIr.

The corresponding discussion in the second paragraph of section 3.2, as well as the other associate discussions in the rest of the manuscript has been revised accordingly.

**Specific:**

P2 L7, why "ratio"?

The term has been changed to 'In terms of the sensitivity of cloud droplet number concentration  $(N_d)$  to  $N_{CCN}$ , the fraction of CCN that converted to cloud droplets  $(N_d/N_{CCN})$  for the weakly (strongly) absorptive regime is 0.69 (0.54)' in the revised manuscript.

P5 L12, Here you imply that the uncertainty in KAZR is 30 m, but it's the uncertainty in KAZR (or MMCR) measurement of cloud top.

Thanks for the correction, the discussions of cloud base and cloud top uncertainty in revised manuscript have been changed to '*The uncertainties of cloud top height detected by MMCR and KAZR are 45 m and 30 m, respectively.*' and '*The laser ceilometer measurement, which is*

sensitive to the second moment, is used to provide an accurate cloud base estimation. The uncertainty of cloud base height is around 10 m (Morris, 2016). Hence, the lidar-radar pair provides the most precise determination of cloud boundaries from a point-based perspective, with combined uncertainties of cloud thickness for MMCR and KAZR periods are 55 m and 40 m, respectively. Note that this will not cause a significant difference in determining the cloud boundaries between these two radar periods.' in the revised manuscript.

P12 L9, I urge you to pay attention to the results of Sena et al. 2016 in ACP who showed how methodological differences in how ACI is calculated can make a large difference.

**Thanks for the suggestion.**

The following sentence: 'A recent study conducted by Sena et al. (2016) within the SGP region showed the different methodologies in calculating ACIr. In particular, different retrieval methods of  $r_e$  could induce large differences.' has been added to the third paragraph of section 3.2 in the revised manuscript.

P19 L16, what rate are you referring to? I think you mean fraction.

**Thanks for the comments.**

The term has been changed to 'In addition, the sensitivity and uncertainty of  $N_d$  are examined in order to estimate the impact of  $N_d$  uncertainty on the assessment of  $N_d/N_{CCN}$ ' in the revised manuscript.

P24 L10, Shouldn't you be referring to Kim et al. here since higher LTS means more adiabatic clouds?

**Thanks for the suggestion.**

The following sentence: 'The result is consistent with the previous study by Kim et al. (2008) who found that  $ACI_r$  is enhanced under adiabatic cloud conditions and higher LTS values are associated with higher cloud adiabaticity' has been added to the third paragraph of section 3.3.6 in the revised manuscript.

P26 L19, Don't you simply mean "ratios". There is no "rate" here that I can see.

The term has been changed to 'The ratios of ...' in the revised manuscript.

**Technical:**

P2 L10, please add definite articles as necessary.P6 L16, Please correct grammar in this section.P20 L24, Please have native speaker check grammar.P27 L9, Paragraph needs grammatical changes.

Thanks for the comments. Dr. Timothy Logan has thoroughly checked and corrected the grammatical issues in the revised manuscript.

[revised manuscript text omitted]
 NCCN values and conversion rates the ratios of Nd to /NCCN in relation to LWP. The conversion rates ratios of Nd/NCCN in the weakly absorptive regime range from 0.58 to 0.86 with a mean value of 0.69, and highly fluctuates with LWP. In contrast, the 5 conversion rates Nd/NCCN in the strongly absorptive regime show lower values and less variability (from 0.47 to 0.64) with a mean value of 0.54. It is interesting to note that the variation of Nd/NCCN in the strongly absorptive regime mimics the variation in NCCN with LWP, indicating a relatively lower aerosol to CCN activating capacity. Therefore, the conversion rate for CCN to cloud dropletNd/NCCN shows no significant dependence on LWP, which is 10 consistent with previous studies, which suggest the response of Nd to the change in NCCN has no fundamental relationship with LWP (e.g., McComiskey et al., 2009). In addition, the sensitivity and uncertainty of Nd isare examined in order to estimate the impact of Nd uncertainty on the assessment of CCN activation rateNd/NCCN. To assess the contributions of different input parameter uncertainties to Nd retrieval, every input parameter was perturbed by 15 its own uncertainty with other parameters held fixed. The results are as follows: (a) an increase (decrease) of LWP by 20  $\text{ gm}^{-2}$  leads to 27.9% (27.6%) change in Nd while an increase (decrease)  $\sigma_x$  by 0.15 leads to a 50.8% (23.9%) change in Nd; (b) an increase (decrease) cloud thickness by 0.15 leads to a 14.5% (23.2%) change in Nd; and (c) an increase (decrease) in re by 10% leads to 14.5% (23.2%) change in Nd. The percentage changes in Nd 
[revised manuscript text omitted]
  |       |     |      |         |    |          |           |         |       |       |     |       |         |

| Data        | Start Time | End Time | A : 6          | Number of   |  |
|-------------|------------|-----------------|----------------|-------------|--|
| Date        | (UTC)      | (UTC)           | Airmass Source | Data Points |  |
| 4 Jan 2007  | 15:00      | 22:30           | S              | 58          |  |
| 5 Jan 2007  | 14:00      | 18:10           | S              | 40          |  |
| 13 Feb 2007 | 17:00      | 22:30           | Ν              | 60          |  |
| 26 Apr 2007 | 14:00      | 17:30           | NE             | 31          |  |
| 21 Nov 2007 | 13:20      | 18:15           | Ν              | 24          |  |
| 14 Feb 2009 | 15:15      | 17:35           | NW             | 29          |  |
| 12 May 2009 | 16:55      | 20:05           | SE             | 37          |  |
| 19 Dec 2009 | 14:40      | 19:35           | NW             | 58          |  |
| 21 Jan 2010 | 15:25      | 22:30           | Ν              | 44          |  |
| 16 Mar 2010 | 15:00      | 20:00           | Ν              | 41          |  |
| 29 Dec 2010 | 16:00      | 18:35           | SE             | 32          |  |
| 26 Mar 2011 | 16:35      | 23:55           | NE             | 59          |  |
| 13 May 2011 | 12:25      | 18:20           | Ν              | 59          |  |
| 4 Feb 2012  | 16:40      | 21:10           | NE             | 37          |  |
| 8 Feb 2012  | 14:30      | 19:45           | Ν              | 54          |  |
| 10 Feb 2012 | 17:15      | 19:50           | NW             | 30          |  |

aAirmass sources denote the relative directions from where the airmasses advected to the ARM-SGP site.

Figure 1. Vertical profiles of liquid water potential temperature ( $\theta_L$ ) and total water mixing ratio (qt) for coupled (a) and decoupled (b) boundary layer conditions. Blue lines denote cloud top and base heights, respectively.